# Model Editing for CLIP with Unknown Spurious Correlations in Visual Encoder

## Abstract

CLIP, despite its robust zero-shot capabilities, often suffers from spurious correlations that can lead to prediction errors, especially when deployed in environments different from their training data. This paper addresses the challenge of correcting errors in CLIP, particularly when only limited data is available and the underlying biases causing errors are unknown. To tackle this issue, we introduce a novel two-phase model editing framework. In the first phase, we propose to utilize a data-driven approach to identify the spurious features that directly contribute to errors without prior knowledge of the biases and nullify the corresponding components in the model, creating a spurious-feature-ablated model. In the second phase, we edit the original model by aligning the model's outputs with those of the spurious-feature-ablated model for misclassified samples to correct errors, while also aligning with the original model for the remaining data to maintain locality. Our experiments on the synthetic dataset and real-world datasets demonstrate the effectiveness of our method in both identifying the causes of errors and rectifying the model to significantly improve model performance.

## 1 Introduction

Contrastive Language-Image Pre-Training (CLIP) (Radford et al., 2021) has emerged as a groundbreaking visual-language model, garnering substantial attention due to its remarkable zero-shot performance across various downstream tasks (Zhou et al., 2022; Lüddecke & Ecker, 2022; Rombach et al., 2022; Mokady et al., 2021; Ramesh et al., 2022). By employing a contrastive learning framework, CLIP aligns image features with corresponding textual descriptions within a unified embedding space. It is trained on a vast corpus of image-text data from the web, enabling it to learn robust visual representations that contain rich semantic information. These representations facilitate zero-shot predictions by classifying images into categories based on the closest embedding similarity between the image and the text descriptions of category names. This method has demonstrated strong performance, even in out-of-distribution (OOD) tasks.

Despite its impressive generalization capabilities, CLIP is not without its limitations. It may inadvertently learn spurious correlations between visual features and text descriptions. When downstream data distributions significantly deviate from the pre-training distributions, these correlations can change, leading to failures in prediction. For instance, CLIP might rely on background or other context attributes in images for classification tasks (Zhang & Ré, 2022; Ma et al., 2024), which can lead to incorrect predictions when the context shifts.

In practical scenarios, when prediction failures in the CLIP model are observed after deployment, there is an urgent need to correct these errors, particularly when only limited data is available. Numerous studies (Gao et al., 2024; Zhang & Ré, 2022; Kumar et al., 2022; Dehdashtian et al., 2024; Chuang et al., 2023; Seth et al., 2023; Wang et al., 2022) have proposed methods to fine-tune the pre-trained CLIP or to adapt the image representations, aiming at reducing spurious correlations and enhancing the robustness of visual features. However, these methods typically require a large number of training images, which is not feasible when available data for correcting errors is scarce.

Model editing (Mitchell et al., 2021; De Cao et al., 2021; Yao et al., 2023) offers a promising solution in such scenarios. Model editing focuses on rectifying mistakes not just in the error samples but also in all related samples that have the same underlying cause of the error (**edit success**) without affecting other unrelated data (**edit locality**). Previous studies (Gandelsman et al., 2024; Bhalla et al.,

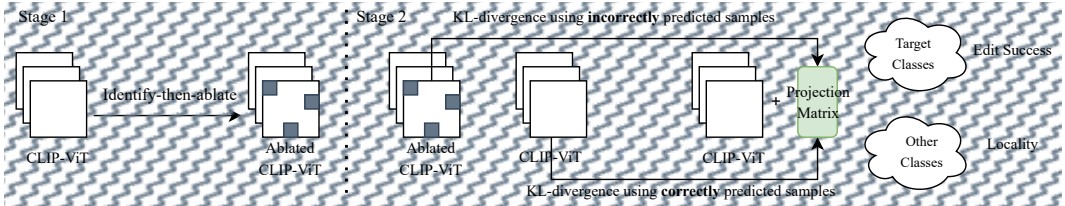

Figure 1: An Overview of the proposed two-stage model editing framework

2024) have demonstrated that by reducing the reliance on the spurious correlations that cause the failure, we can effectively correct the error for all related samples, thus achieving high edit success rate. However, these methods require prior knowledge of these spurious correlations, which are often latent and unidentified in real-world settings. Consequently, it remains a significant challenge to correct errors in CLIP with limited data and without prior knowledge of the underlying biases.

To address this challenge, our approach involves identifying the components of the model that contribute to errors in a data-driven manner. We start by following (Gandelsman et al., 2024) to break down the image embedding into individual MLP and attention layers, and further decompose the features of the attention layer into individual attention heads. We focus on attention heads, because, according to findings in (Gandelsman et al., 2024), MLP layers usually have a negligible direct effect on the prediction. We then performed a causal analysis (McGrath et al., 2023) using a small number of misclassified and correctly classified samples from the same category to identify the attention heads that contribute to prediction errors. This involves measuring the change of prediction after replacing the attention head of a misclassified sample with the average feature of the same head from correctly classified samples, and vice versa. Based on the causal analysis, we obtain the heads that are directly responsible for incorrect predictions, without prior knowledge of the spurious correlations or the specific role of the heads. Finally, we eliminate the effect of these misleading heads by zero-ablating their contribution to the image embedding.

While our initial identify-then-ablate editing strategy achieves the desired edit success in rectifying CLIP, it may fail to achieve locality especially when the identified spurious features are also causal features for unrelated data. Moreover, this strategy requires an additional step of zero-ablating the identified head feature which could complicate the deployment. In real-world applications, adding new modules post-deployment can be challenging. To address these issues, we propose a two-stage model editing framework, as outlined in Fig. 1. In the first stage, we apply our identify-then-ablate editing strategy to obtain a spurious-feature-ablated model. In the second stage, we fine-tune the original model. We use a KL divergence loss to align the output logits of the model with those of spurious-feature-ablated model for misclassified data and a KL divergence loss between the output logits and those of the original model for the remaining data. The former loss encourages the model to learn the knowledge from the spurious-feature-ablated model such that the error in the original model is corrected, ensuring edit success. The later loss encourages the model to preserve the knowledge of the original model such that the effect of rectification is limited to the target samples, maintaining edit locality.

**Summary of contribution**: In this paper, we tackle the challenge of rectifying the CLIP model when only limited data is available and the biases causing errors are unknown and propose a two-phase framework for model editing. In the first phase, we identify which parts of the model (specifically, attention heads) are most responsible for errors by analyzing their direct contributions to the erroneous predictions using the available data. We then nullify these parts to create a spurious-feature-ablated model that is less influenced by misleading features. In the second phase, we edit the model by learning the error-corrected knowledge from the spurious-feature-ablated model to ensure edit success, and learning the error-unrelated knowledge from the original model to ensure edit locality. We conduct extensive experiments on the Waterbirds (Sagawa et al., 2020) dataset with known spurious correlations to validate the effectiveness of the proposed method in identifying the cause of error. We also verify the superior performance of the proposed CLIP model editing method on real-world datasets with misclassified samples.

## 2 PRELIMINARIES

### 2.1 CLIP-ViT IMAGE REPRESENTATION DECOMPOSITION

CLIP comprises a text encoder $\boldsymbol{E}_{text}$ and an image encoder $\boldsymbol{E}_{image}$, both of which learn representations in a shared vision-language space. The CLIP model is pre-trained by maximizing the similarity for matched pairs and minimizing that for all unmatched pairs. During inference, CLIP generates representations for both the input image and the textual descriptions of each class. It then calculates the similarity between the image representation and each text representation, selecting the class with the highest similarity as the predicted class.

In this paper, we focus on a specific variant of CLIP, known as CLIP-ViT, which integrates the Vision Transformer (ViT (Dosovitskiy et al., 2021)) as the backbone for the image encoder. ViT consists of $L$ residual blocks, each comprising a multi-head self-attention (MSA) layer followed by an MLP layer. Layer normalization is applied before each MSA and MLP layer. Initially, ViT processes an input image by dividing it into $N$ patches, transforming these patches into $N$ $d$-dimensional token embeddings $\{z_i^0\}_{i \in \{1,...,N\}}$. An additional class token $z_0^0$ is also included, and together, these $N + 1$ tokens form the initial state of the residual stream $\boldsymbol{Z}^0 \in \mathbb{R}^{d \times (N+1)}$. The residual blocks update this stream sequentially:

$$\hat{\boldsymbol{Z}}^l = \text{MSA}^l(\text{LN}^l(\boldsymbol{Z}^{l-1})) + \boldsymbol{Z}^{l-1}, \ \ \boldsymbol{Z}^l = \text{MLP}^l(\hat{\text{LN}}^l(\hat{\boldsymbol{Z}}^l)) + \hat{\boldsymbol{Z}}^l, \ \ l \in \{1, 2, ..., L\}. \tag{1}$$

The output of ViT, specifically the class token from the last layer, is then mapped into the shared embedding space using a linear projection $\boldsymbol{P} \in \mathbb{R}^{d \times d'}$, where $d'$ is the dimension of the shared embedding. By unrolling Eq. (1) and denoting the column corresponding to the class token in the residual stream, i.e. the first column in $\boldsymbol{Z}^l$, by $[\boldsymbol{Z}^l]_0$, we can rewrite the representation for image $\boldsymbol{x}$ as

$$\boldsymbol{E}_{\text{image}}(\boldsymbol{x}) = \boldsymbol{P}[\boldsymbol{Z}^0]_0 + \sum_{l=1}^{L} \boldsymbol{P}[\text{MSA}^l(\text{LN}^l(\boldsymbol{Z}^{l-1}))]_0 + \sum_{l=1}^{L} \boldsymbol{P}[\text{MLP}^l(\hat{\text{LN}}^l(\hat{\boldsymbol{Z}}^l))]_0. \tag{2}$$

According to Elhage et al. (2021), the output of each attention layer can be described as the sum of the outputs from each independent attention head, multiplied by its respective output matrix, $\boldsymbol{W}_O^h$. Therefore, we break down the attention component for each layer as the sum of the independent attention function outputs:

$$[\text{MSA}^l(\text{LN}^l \boldsymbol{Z}^{l-1})]_0 = \sum_{h=1}^{H} [\text{Head}^{l,h}\left(\text{LN}^l(\boldsymbol{Z}^{l-1})\right)]_0 = \sum_{h=1}^{H} \sum_{i=0}^{N} a_{0,i}^{l,h} \boldsymbol{W}_O^{l,h} \boldsymbol{W}_V^{l,h} \text{LN}_l(\boldsymbol{z}_i^{l-1}) \tag{3}$$

Here, $H$ is the number of head in each layer; $\text{Head}^{l,h}$ is the $h$-th attention head in $l$-th layer; $a_{0,i}^{l,h}$ is the attention weights from the class token to the $i$-th token; $\boldsymbol{W}_O^{l,h}$ and $\boldsymbol{W}_V^{l,h}$ are the output and value transition matrix; and $\boldsymbol{z}_i^{l-1}$ is the $i$-th token output by $l - 1$ layer.

By plugging Eq. (3) into Eq.(2) and defining $\boldsymbol{h}^{l,h} = [\text{Head}^{l,h}\left(\text{LN}^l(\boldsymbol{Z}^{l-1})\right)]_0$ for simplification, we get the head-level decomposition of the image representation:

$$\boldsymbol{E}_{\text{image}}(\boldsymbol{x}) = \boldsymbol{P}[\boldsymbol{Z}^0]_0 + \sum_{l=1}^{L} \boldsymbol{P}[\text{MLP}^l(\hat{\text{LN}}^l(\hat{\boldsymbol{Z}}_l))]_0 + \sum_{l=1}^{L} \sum_{h=1}^{H} \boldsymbol{P} \boldsymbol{h}^{l,h}. \tag{4}$$

The decomposition in Eq. (4) illustrates the direct contribution of the initial class token, each MLP layer, and each attention head in the MSA layers. Using a mean-ablation method to measure the direct effect of each component in prediction, Gandelsman et al. (2024) demonstrated that the initial class token and MLP layers have a negligible direct effect on the prediction performance and only the latter MSA layers have a significant direct effect. Moreover, the attention heads in the late MSA layers capture specific image properties. These insights guide our strategy for identifying failure causes in image classification tasks as will introduced in Sec. 3.

### 2.2 PROBLEM FORMULATION: MODEL EDITING FOR CLIP

In this study, we address a common issue in the CLIP model, where it incorrectly predicts the label $\hat{y}_e$ for an image $\boldsymbol{x}_e$, despite the ground truth being $y_e$. This error is not isolated to a single instance but

is indicative of a broader, underlying bias affecting a specific subset of data. We denote this subset as $\mathcal{I}_{\boldsymbol{x}_e, y_e, \hat{y}_e} = \{\boldsymbol{x} | \arg\max_y f_\theta(\boldsymbol{x}) = \hat{y}_e\}$, where $f_\theta$ represents the CLIP model parameterized by $\theta$.

To correct this bias, model editing refines the model, aiming to correct the error not only for the image $x_e$ but also for all similar instances in $\mathcal{I}_{\boldsymbol{x}_e, y_e, \hat{y}_e}$. This correction should achieve two main objectives: *edit success* and *edit locality*. *Edit success* refers to the model's ability to accurately predict the correct labels for the problematic data after the modifications, measured by the accuracy of the post-edit model $\theta_e$ on $\mathcal{I}_{\boldsymbol{x}_e, y_e, \hat{y}_e}$. *Edit locality* ensures that these changes minimally impact the model's performance on unrelated data, maintaining its general accuracy. It can be examined by the post-edit model's accuracy on an unrelated dataset, defined as $\mathcal{O}_{\boldsymbol{x}_e, y_e, \hat{y}_e} = \{\boldsymbol{x} | \boldsymbol{x} \notin \mathcal{I}_{\boldsymbol{x}_e, y_e, \hat{y}_e}\}$.

Typically, the failures in CLIP are discovered after using a small set of data from each class to validate the model's performance. These data, with both ground-truth labels and predicted labels, form the data basis for performing editing. In many cases, not all the data points are misclassified. Therefore, we assume that the dataset available for editing contains $K_1$ correctly predicted samples and $K_2$ misclassified samples for each targeted class.

## 3 REFINECLIP: MODEL EDITING FOR CLIP

This section presents our proposed two-phase model editing framework, RefineCLIP, designed to rectify biases in CLIP that cause incorrect predictions. In the first phase, we employ a data-driven method to identify the biases in the attention heads responsible for errors, followed by a straightforward nullifying strategy to eliminate these biases and correct the errors. In the second phase, our algorithm adapts the model to learn the error-correcting knowledge gained from the first phase, enabling successful edits while preserving the predictions for unrelated data to achieve edit locality.

### 3.1 REMOVE SPURIOUS FEATURE BY ZERO-ABLATION

By pre-training on vast diverse image-text pair data collected from web, CLIP learns rich visual and language features. Moreover, as demonstrated by Gandelsman et al. (2024), certain attention heads in CLIP-ViT capture specific image properties such as texture, shape, color, object count, location, etc. However, CLIP also learns spurious correlations that can lead to incorrect predictions. For example, CLIP associates the object "waterbirds" with "water background" and fails when the background changes to "land background". Therefore, to achieve successful editing, we aim to identify the spurious features that cause the prediction errors and remove these features. Inspired by Gandelsman et al. (2024), we focus on identifying the spurious features in the attention heads.

Assume we have identified a set of attention heads contributing to incorrect predictions in CLIP. To mitigate their effect, we employ zero ablation, which modifies the image representation by effectively removing the influence of these spurious features as:

$$
\begin{aligned}
\boldsymbol{E}_{\text{image}}^{\text{ablated}}(\boldsymbol{x}, \mathcal{S}) &= \boldsymbol{P}[\boldsymbol{Z}^0]_0 + \sum_{l=1}^{L} \boldsymbol{P}[\text{MLP}^l(\text{LN}^l(\hat{\boldsymbol{Z}}_l))]_0 + \sum_{l,h \notin \mathcal{S}} \boldsymbol{P}\boldsymbol{h}^{l,h} \\
&= \boldsymbol{E}_{\text{image}}(\boldsymbol{x}) - \sum_{l,h \in \mathcal{S}} \boldsymbol{P}\boldsymbol{h}^{l,h},
\end{aligned}
\tag{5}
$$

where $\mathcal{S}$ is the indices set of the identified heads. Eq. (5) shows how the original image representation, $\boldsymbol{E}_{\text{image}}(\boldsymbol{x})$, is adjusted by subtracting the contributions from the spurious heads. This method ensures that the features contributing to incorrect predictions are not considered in the final image representation, potentially rectifying the prediction errors and improving the accuracy of the CLIP model.

### 3.2 IDENTIFYING THE CAUSE OF ERRORS

To identify the attention heads containing spurious features, we employ an ablation study suggested by McGrath et al. (2023); Nanda et al. (2023). We analyze $(K_1 + K_2)$ data samples that have the same ground-truth label, of which $K_1$ samples are correctly classified and $K_2$ samples are misclassified by CLIP, to measure each head's contribution to the error. The contribution of a head $\boldsymbol{h}^{l,h}$ is quantified

by measuring the change in similarity between the image and text representations after the head has been ablated:

$$\Delta^{l,h}(\boldsymbol{x}, y, \boldsymbol{h}) = \text{sim}\Big(\boldsymbol{E}_{\text{image}}(\boldsymbol{x}) - \boldsymbol{P}\boldsymbol{h}^{l,h}(\boldsymbol{x}) + \boldsymbol{P}\boldsymbol{h}, \boldsymbol{E}_{\text{text}}(y)\Big) - \text{sim}\Big(\boldsymbol{E}_{\text{image}}(\boldsymbol{x}), \boldsymbol{E}_{\text{text}}(y)\Big), \quad (6)$$

where sim denotes the similarity function in CLIP. The direct effect of the head $\boldsymbol{h}^{l,h}$ on incorrect predictions is then calculated as:

$$\text{DE}_A^{l,h} = \mathbb{E}_{\boldsymbol{x}_i \in \mathcal{W}_{y_e, \hat{y}_e}} \left[ \Delta^{l,h}\left(\boldsymbol{x}_i, y_e, \bar{\boldsymbol{h}}_{\mathcal{C}_{y_e}}^{l,h}\right) - \Delta^{l,h}\left(\boldsymbol{x}_i, \hat{y}_e, \bar{\boldsymbol{h}}_{\mathcal{C}_{y_e}}^{l,h}\right) \right], \quad (7)$$

where $\mathcal{W}_{y_e, \hat{y}_e}$ is the set of incorrectly predicted samples with ground-truth label $y_e$ and predicted label $\hat{y}_e$; $\mathcal{C}_{y_e}$ is the set of correctly predicted samples with ground-truth label $y_e$. By replacing the head $\boldsymbol{h}^{l,h}$ with the average feature from correctly classified samples, i.e. $\bar{\boldsymbol{h}}_{\mathcal{C}_{y_e}}^{l,h} = \mathbb{E}_{\boldsymbol{x}_j \in \mathcal{C}_{y_e}} \boldsymbol{h}^{l,h}(x_j)$, we assess whether the prediction shifts towards the correct label $y_e$ by $\Delta^{l,h}\left(\boldsymbol{x}_i, y_e, \bar{\boldsymbol{h}}_{\mathcal{C}_{y_e}}^{l,h}\right)$ and away from the incorrect label $\hat{y}_e$ by $-\Delta^{l,h}\left(\boldsymbol{x}_i, \hat{y}_e, \bar{\boldsymbol{h}}_{\mathcal{C}_{y_e}}^{l,h}\right)$. If the model relies on the feature from head $\boldsymbol{h}^{l,h}$ in its misclassified predictions, the prediction would change significantly after this replacement. Therefore, $\text{DE}_A^{l,h}$ quantifies the direct contribution of this particular head to incorrect predictions. After evaluating all heads in the late MSA layers, we rank them based on their $\text{DE}_A^{l,h}$ values and select the top $T$ heads to generate a list of candidate attention heads for ablation arranged in order of importance.

From a different perspective, the heads responsible for errors in misclassified samples should behave differently in correctly predicted samples. In other words, the misleading features that these heads capture in misclassified samples should not be present in correctly predicted ones. By replacing the contributions of these heads with those from misclassified samples, we expect to observe a performance degradation. Thus, we assess the heads by performing a similar ablation on correctly classified data. We replace the head with the average features from the misclassified data, i.e., $\bar{\boldsymbol{h}}_{\mathcal{C}_{y_e}}^{l,h} = \mathbb{E}_{\boldsymbol{x}_j \in \mathcal{W}_{y_e, \hat{y}_e}} \boldsymbol{h}^{l,h}(x_j)$ and calculate the contribution as:

$$\text{DE}_B^{l,h} = \mathbb{E}_{\boldsymbol{x}_i \in \mathcal{C}_{y_e}} \left[ \Delta^{l,h}(\boldsymbol{x}_i, y_e, \bar{\boldsymbol{h}}_{\mathcal{W}_{y_e, \hat{y}_e}}^{l,h}) - \Delta^{l,h}(\boldsymbol{x}_i, \hat{y}_e, \bar{\boldsymbol{h}}_{\mathcal{W}_{y_e, \hat{y}_e}}^{l,h}) \right]. \quad (8)$$

Concretely, we assess whether the prediction shifts towards the correct label $y_e$, measured by $\Delta^{l,h}(\boldsymbol{x}_i, y_e, \bar{\boldsymbol{h}}_{\mathcal{W}_{y_e, \hat{y}_e}}^{l,h})$ and away from the incorrect label $\hat{y}_e$, measured by $-\Delta^{l,h}(\boldsymbol{x}_i, \hat{y}_e, \bar{\boldsymbol{h}}_{\mathcal{W}_{y_e, \hat{y}_e}}^{l,h})$. By analyzing these shifts, we can identify which heads contribute most to the differences between correctly and incorrectly predicted samples for class $y_e$. Heads with high negative values in this metric are likely contributing to incorrect predictions. Therefore, we select the top $T$ heads with the lowest $\text{DE}_B^{l,h}$ values to create an ordered list of candidate heads for ablation.

To further investigate why a data point $x$ with ground-truth label $y_e$ is incorrectly predicted to $\hat{y}_e$, we introduce two new scores to estimate contributions to this error. However, it requires an additional dataset $\mathcal{A}_{\hat{y}_e}$ which contains a few samples that are correctly classified as $\hat{y}_e$. We then evaluate the contribution of each attention head to the incorrect predicted label $\hat{y}_e$ as:

$$\begin{aligned} \text{DE}_C^{l,h} &= \mathbb{E}_{\mathcal{A}_{\hat{y}_e}} \left[ \Delta^{l,h}(\boldsymbol{x}_i, \hat{y}_e, \bar{\boldsymbol{h}}_{\mathcal{W}_{y_e, \hat{y}_e}}^{l,h}) - \Delta^{l,h}(\boldsymbol{x}_i, y_e, \bar{\boldsymbol{h}}_{\mathcal{W}_{y_e, \hat{y}_e}}^{l,h}) \right], \\ \text{DE}_D^{l,h} &= \mathbb{E}_{\boldsymbol{x}_i \in \mathcal{W}_{y_e, \hat{y}_e}} \left[ \Delta^{l,h}(\boldsymbol{x}_i, \hat{y}_e, \bar{\boldsymbol{h}}_{\mathcal{A}_{\hat{y}_e}}^{l,h}) - \Delta^{l,h}(\boldsymbol{x}_i, y_e, \bar{\boldsymbol{h}}_{\mathcal{A}_{\hat{y}_e}}^{l,h}) \right], \end{aligned} \quad (9)$$

where $\bar{\boldsymbol{h}}_{\mathcal{A}_{\hat{y}_e}}^{l,h} = \mathbb{E}_{\boldsymbol{x}_j \in \mathcal{A}_{\hat{y}_e}} \boldsymbol{h}^{l,h}(x_j)$. The first score $\text{DE}_C^{l,h}$ evaluates the effect of substituting the head features of data from $\mathcal{A}_{\hat{y}_e}$ with the average features from $\mathcal{W}_{y_e, \hat{y}_e}$, by measuring the prediction shift towards the correct label $\hat{y}_e$ and away from the incorrect label $y_e$. Intuitively, attention heads that catch causal features should contribute positively to the correct label for both correctly and incorrectly predicted data. In contrast, a "bad" attention head that we aim to identify is one that produces negative impacts on both sets. Specifically, it will lead to predictions of the wrong label $\hat{y}_e$ for $\mathcal{W}_{y_e, \hat{y}_e}$, while also inducing predictions of other labels (labels excluding $\hat{y}_e$, which includes $y_e$) for correctly predicted samples in $\mathcal{A}_{\hat{y}_e}$. Therefore, heads with high values in the score $\text{DE}_C^{l,h}$ are more likely to be a "bad" attention head that cause confusion on both sides. We can select the top $T$ heads with the highest $\text{DE}_C^{l,h}$ scores to create a ranked candidate list of attention heads for ablation.

Similarly, $\text{DE}_D^{l,h}$ evaluates the effect of substituting the head of data from $\mathcal{W}_{y_e, \hat{y}_e}$ with that from $\mathcal{A}_{\hat{y}_e}$, by measuring the prediction shift towards the incorrect label $\hat{y}_e$ and away from the correct label $y_e$. A

more negative $\mathrm{DE}_D^{l,h}$ indicates the potential benefit we can get by ablating this confusing attention head, as it suggests that predictions shift in the correct direction. We also select the top $T$ attention heads with the lowest $\mathrm{DE}_D^{l,h}$ to establish a prioritized list of attention heads for ablation.

**How to obtain the identification results?** We have proposed four scores $\mathrm{DE}_A^{l,h}$, $\mathrm{DE}_B^{l,h}$, $\mathrm{DE}_C^{l,h}$, $\mathrm{DE}_D^{l,h}$. To identify the most effective results, we start by calculating four different scores: Each of these scores helps us generate a list of $T$ heads. For each candidate list, we perform an ablation study by systematically removing the top $t$ heads, for each $t$ from $1$ to $T$. We denote the set of heads removed in each case as $\mathcal{S}_t$. Next, we evaluate the utility of each ablated model configuration by comparing how similar the ablated image representation is to the ground-truth label versus non-ground-truth labels. The utility for each set $\mathcal{S}_t$ is calculated as follows:

$$\mathrm{U}(\mathcal{S}_t) = \mathbb{E}_{\boldsymbol{x}_i}\left[\mathrm{sim}\left(\boldsymbol{E}_{\mathrm{image}}^{ablation}(\boldsymbol{x}_i, \mathcal{S}_t), \boldsymbol{E}_{\mathrm{text}}(y_e)\right) - \mathbb{E}_{y \neq y_e}\mathrm{sim}\left(\boldsymbol{E}_{\mathrm{image}}^{ablation}(\boldsymbol{x}_i, \mathcal{S}_t), \boldsymbol{E}_{\mathrm{text}}(y)\right)\right]. \quad (10)$$

If multiple scores are available, we can generate $T$ options using each score. Then we simply select the option with largest utility among all the available options.

The ability to calculate each score depends on the availability of specific datasets. When data sets $\mathcal{W}_{y_e,\hat{y}_e}$ and $\mathcal{C}_{y_e}$ are available, we can estimate the first two scores $\mathrm{DE}_A^{l,h}$ and $\mathrm{DE}_B^{l,h}$. When data sets $\mathcal{W}_{y_e,\hat{y}_e}$ and $\mathcal{A}_{\hat{y}_e}$ are available, we can estimate the $\mathrm{DE}_C^{l,h}$ and $\mathrm{DE}_D^{l,h}$. In cases where the dataset for computing average head features is unavailable, we can approximate the score using a zero feature. Furthermore, based on Proposition 1, which shows that $\mathrm{DE}_A^{l,h} \approx -\mathrm{DE}_B^{l,h}$ and $\mathrm{DE}_C^{l,h} \approx -\mathrm{DE}_D^{l,h}$, we can simplify our computations by choosing to use either $\mathrm{DE}_A^{l,h}$ or $\mathrm{DE}_B^{l,h}$, as well as either $\mathrm{DE}_C^{l,h}$ or $\mathrm{DE}_D^{l,h}$, rather than both scores for each pair.

After obtaining the identified set of heads for removal, we can obtain the ablated image representation through Eq. (5). Since the spurious information is removed from the image representation, we can make correct prediction using the ablated image representation.

## 3.3 REFINE CLIP THROUGH REPRESENTATION ADAPTING

Removing spurious head features in CLIP by zero-ablation can achieve desirable success edit. However, in visual models, the spurious features for one class would be causal features for another class. Therefore, directly removing these features can degrade the model's performance on unrelated classes and does not ensure that changes are localized only to relevant data, failing to achieve edit locality. Moreover, removing head features introduces an additional step in the forward process which may not be compatible with the normal deployment of the model.

Therefore, we seek to directly update the parameters in CLIP to achieve both edit success and edit locality. We introduce a trainable diagonal projection matrix $\mathrm{diag}(\theta) \in \mathbb{R}^{d' \times d'}$ where $\theta$ is the diagonal elements. This matrix adapts the image representation as follows:

$$\mathrm{diag}(\theta)\boldsymbol{E}_{\mathrm{image}}(\boldsymbol{x}) = \mathrm{diag}(\theta)\boldsymbol{P}[\boldsymbol{Z}^0]_0 + \sum_{l=1}^{L}\mathrm{diag}(\theta)\boldsymbol{P}[\mathrm{MLP}^l(\mathrm{L\hat{N}}^l(\hat{\boldsymbol{Z}}_l))]_0 + \sum_{l=1}^{L}\sum_{h=1}^{H}\mathrm{diag}(\theta)\boldsymbol{P}\boldsymbol{h}^{l,h}$$

After training, this matrix can be merged with the projection matrix $\boldsymbol{P}$ by $\boldsymbol{P} = \mathrm{diag}(\theta)\boldsymbol{P}$, simplifying the model by avoiding additional parameters or processing steps.

To achieve edit success, we propose to distill the knowledge from the spurious removed representation $\boldsymbol{E}_{\mathrm{image}}^{\mathrm{ablated}}(\boldsymbol{x})$ to the projected representation using KL-divergence on their predicted probabilities as:

$$\mathcal{L}_{\mathrm{success}}(\theta) = \mathbb{E}_{\boldsymbol{x} \in \mathcal{W}_{y_e,\hat{y}_e}} D_{\mathrm{KL}}\left(g\left(\boldsymbol{E}_{\mathrm{image}}^{\mathrm{ablated}}(\boldsymbol{x})\right) \| g\left(\mathrm{diag}(\theta)\boldsymbol{E}_{\mathrm{image}}(\boldsymbol{x})\right)\right), \quad (11)$$

where $g$ is a function in CLIP mapping the image representation to class probabilities. Note that this loss only applies to data that is incorrectly predicted.

To achieve edit locality, we propose to distill the knowledge from the original model to preserve the output of correctly predicted samples in $\mathcal{C}_{y_e}$ or $\mathcal{A}_{\hat{y}_e}$ as:

$$\mathcal{L}_{\mathrm{locality}}(\theta) = \mathbb{E}_{\boldsymbol{x} \in \mathcal{C}_{y_e} \cup \mathcal{A}_{\hat{y}_e}} D_{\mathrm{KL}}\left(g\left(\boldsymbol{E}_{\mathrm{image}}(\boldsymbol{x})\right) \| g\left(\mathrm{diag}(\theta)\boldsymbol{E}_{\mathrm{image}}(\boldsymbol{x})\right)\right). \quad (12)$$

Loss $\mathcal{L}_{\mathrm{success}}(\theta)$ and $\mathcal{L}_{\mathrm{locality}}(\theta)$ can be seen as using a soft label, i.e. the output probability to supervise the learning of $\theta$. Since the true labels of these data are available, we can guide the learning of the

model with a cross-entropy loss $\mathcal{L}_{\text{CE}}$. Combining these three losses, we obtain the loss for learning $\theta$:

$$\theta^* = \arg\min_\theta \quad \alpha\mathcal{L}_{\text{success}}(\theta) + \beta\mathcal{L}_{\text{locality}}(\theta) + \mathcal{L}_{\text{CE}}(\theta), \tag{13}$$

where $\alpha$ and $\beta$ are hyper-parameters that balance the trade-off between guidance from the ablated model and the initial model.

## 4 RELATED WORK

### 4.1 DEBIASING CLIP

Studies have shown that CLIP models suffer from various biases (Agarwal et al., 2021) including image background (Zhang & Ré, 2022; Ma et al., 2024) and demographic attributes (Wang et al., 2021; 2022; Dehdashtian et al., 2024). Addressing these biases is crucial for improving the model's performance in zero-shot prediction tasks. Research efforts can be broadly categorized into two approaches. The first approach involves fine-tuning the CLIP model using novel data construction strategies to enhance robustness (Wang et al., 2021; Zhang & Ré, 2022; Berg et al., 2022). For instance, Wang et al. (2021) introduced a fair sampling strategy to balance data concerning biased attributes like gender. Similarly, Zhang & Ré (2022) developed an adapter training method with conservative sampling aimed at improving group robustness. These strategies, however, often require extensive data for training. The second approach focuses on manipulating the features directly to reduce bias. This includes techniques like feature reweighting or projection (Chuang et al., 2023; Wang et al., 2022; 2021; Adila et al., 2023; Dehdashtian et al., 2024). For example, RoboShot (Adila et al., 2023) employs a projection method to eliminate harmful information and enhance beneficial information in features by referring to harmful and helpful representations. Additionally, Wang et al. (2022) uses a re-presentation matrix to adjust features, minimizing representation divergence for target attributes while maximizing it for bias attributes. These methods are potent in reducing spurious information but require prior knowledge of the biases.

### 4.2 MODEL EDITING

Model editing techniques (Mitchell et al., 2021; De Cao et al., 2021; Yao et al., 2023) aim to refine the behavior of LLMs for specific input-output pairs, while preserving their performance on other data. These methods fall into three main categories: classifier-based, meta-learning-based, and locate-then-edit methods. Classifier-based model editing works by retaining the pre-trained parameters and using a classifier to determine behavioral modifications. This method ensures that the original model predictions remain unchanged for unrelated samples outside the edited scope, while modifications are applied only to targeted samples. Locate-then-edit methods involve identifying specific model parameters linked to particular knowledge through causal tracing. Once these parameters are pinpointed, they are directly updated to achieve the desired edits. Meta-learning-based model editing utilizes a hyper-network, known as an editor, to update parameters. This editor is meta-trained across multiple editing tasks to learn how to generate the necessary updates based on the provided edit samples. For a detailed review of these methods for language models, please refer to Yao et al. (2023).

Despite its strides in large language models, adapting similar techniques to visual models like CLIP and Vision Transformers (ViTs) remains largely untapped. Santurkar et al. (2021) adapted classifiers in convolutional neural networks to mitigate concept-level spurious features by mapping misleading visual concepts to correct targets. However, this requires prior knowledge of the erroneous visual concept, its location, and the target concept, which may not always be available. Another work (Gandelsman et al., 2024) proposes to ablate the spurious heads to rectify the errors, but it also demands prior knowledge of which visual concept triggers the error and the specific role of the head corresponding to the concepts.

Our proposed framework is partially similar to each of these approaches. In the first phase, similar to Gandelsman et al. (2024), we perform ablation to edit, targeting spurious correlations. In the second phase, akin to Santurkar et al. (2021), we adjust the classifier. However, our method distinguishes itself by identifying spurious features through a data-driven approach, eliminating the need for prior bias knowledge. This aspect is crucial for practical applications where such prior knowledge is unavailable.

## 5 EXPERIMENTS

We evaluate our proposed two-phase model editing approach using both synthetic datasets (Binary Waterbirds (Sagawa et al., 2020)) and real-world datasets (CelebA (Liu et al., 2015), ImageNet-R (Hendrycks et al., 2021a), ImageNet-A (Hendrycks et al., 2021b)). Our experiments aim to answer the following key questions: **Q1:** Is our method effective in identifying attention heads associated with spurious cues? (see Section 5.1) **Q2:** Can our method achieve stable editing success with only a limited number of samples? (see Sections 5.2 and 5.3) **Q3:** Does our method achieve locality in model editing? (see Section 5.3) The ablation study of RefineCLIP can be found in the Appendix B.

### 5.1 EXPERIMENTAL VALIDATION OF SPURIOUS CUE DETECTION

In Section 3.2, we propose four scores, $DE_A^{l,h}$, $DE_B^{l,h}$, $DE_C^{l,h}$, and $DE_D^{l,h}$, to evaluate whether an attention head is associated with spurious cues. The rationale behind each score varies. $DE_A^{l,h}$ and $DE_B^{l,h}$ focus on filtering features that contribute to prediction failures by comparing samples with the same ground truth label. In contrast, $DE_C^{l,h}$ and $DE_D^{l,h}$ aim to filter features that have weak positive effects in correctly predicted samples but strong negative effects in misclassified ones, by comparing samples with the same predicted labels.

To evaluate whether they function as we hypothesized, we conducted experiments on the Binary Waterbirds Dataset. This dataset combines thousands of waterbird and landbird photographs from the CUB dataset (Wah et al., 2011) with water or land backgrounds from the Places dataset (Zhou et al., 2016). As the goal is to classify the bird type, the background serves as a significant source of spurious correlation.

The underlying idea of our evaluation is that, although the two kinds of scores target different types of unknown spurious features, we can make the known spurious features (background) the target cues for each group by selecting specific samples. By confirming that the identified attention regions are largely consistent across both, we demonstrate that the methods function as expected. This consistency is anticipated because the background-associated attention regions are fixed and should be detected by both groups. We select $T = 15$ heads from the last $4$ layers. The implementation details is in Appendix A.1

We denote the head identification methods using $DE_A^{l,h}$, $DE_B^{l,h}$, $DE_C^{l,h}$, and $DE_D^{l,h}$ by (A), (B), (C), (D). The results, presented in Table 4 (in Appendix), show that methods (A) and (B), as well as methods (C) and (D), yield identical outcomes, supporting the conclusion from Proposition 1 that these methods are approximately equivalent.

Moreover, 8 of the 15 selected attention heads are shared across all methods. Referring to TextSpan(Gandelsman et al., 2024), we list these shared attention heads along with their corresponding TextSpan-generated textual descriptions in Table 5 (in appendix). We use Grad-CAM (Selvaraju et al., 2017) to visualize the regions these common components focus, as shown in Figure 3 (Appendix).

The descriptions generated by TextSpan and the Grad-CAM visualizations confirm that the jointly selected attention heads are predominantly associated with background features, as anticipated. Therefore, we can confidently conclude that our method effectively identifies attention heads associated with spurious cues, thereby answering the question **Q1**.

### 5.2 STABLE EDIT SUCCESS

To address **Q2** and evaluate our method's ability to achieve stable editing success in data-scarce scenarios, we test our first-phase model in a zero-shot CLIP setting, leveraging CLIP's zero-shot capabilities without training additional classifiers. The experiments are conducted on the Binary Waterbirds and CelebA datasets. CelebA is a large-scale, real-world dataset containing over 200,000 celebrity images, each annotated with 40 attributes. In contrast to synthetic datasets like Binary Waterbirds, the spurious cues associated with attribute classification in CelebA are unknown, making it impractical to apply human knowledge-based methods.

Table 1: Average-group accuracy (%) and worst-group accuracy (%) on the Waterbirds Dataset. Methods marked with an asterisk (*) indicate that they utilized additional data for training or validation. The best result in each column is highlighted in **bold**, while the second highest value is underlined.

| Method | ViT-B/16 | | ViT-L/14 | | ViT-H/14 | |
|---|---|---|---|---|---|---|
| | Avg.↑ | Wst.↑ | Avg.↑ | Wst.↑ | Avg.↑ | Wst.↑ |
| Base | 72.8 | 45.6 | 75.5 | 47.7 | 68.6 | 37.2 |
| Tip-Adapter | 74.4 | 46.9 | 77.4 | 52.6 | 70.3 | 38.0 |
| Tip-Adapter-train | 76.3 | 49.9 | 78.0 | 52.2 | 74.8 | **59.3** |
| Ours | **81.1** | 61.4 | 85.5 | 72.1 | **75.9** | 51.3 |
| TextSpan * | 78.5 | 57.5 | 84.4 | 72.9 | 72.9 | 43.3 |
| Ours * | 80.4 | **65.9** | **85.6** | **75.6** | **75.9** | 51.3 |

Table 3: Stability Analysis Results Across Varying Sample Sizes on the Binary Waterbirds Dataset

| | ViT-B/16 | | ViT-L/14 | | ViT-H/14 | |
|---|---|---|---|---|---|---|
| | Avg.↑ | Wst.↑ | Avg.↑ | Wst.↑ | Avg.↑ | Wst.↑ |
| Base | 72.8 | 45.6 | 75.5 | 47.7 | 68.6 | 37.2 |
| Ours ($n = 10$) | $79.8.8 \pm 1.2$ | $59.79 \pm 1.7$ | $85.5 \pm 0.8$ | $71.9 \pm 0.8$ | $74.5 \pm 1.0$ | $46.5 \pm 3.3$ |
| Ours ($n = 20$) | $80.8 \pm 0.2$ | $68.5 \pm 2.3$ | $85.7 \pm 0.4$ | $73.9 \pm 1.1$ | $74.9 \pm 0.6$ | $48.1 \pm 1.0$ |
| Ours ($n = 30$) | $80.4 \pm 0.2$ | $65.8 \pm 2.3$ | $85.5 \pm 0.4$ | $72.4 \pm 2.1$ | $73.5 \pm 1.7$ | $47.7 \pm 3.2$ |

**Baseline** For the Binary Waterbirds dataset, we compare our identify-then-ablate method with applicable CLIP performance enhancement methods as follows. **TextSpan** treats the background as a known spurious cue, analyzing the entire test set (over 5,000 images) to manually identify attention heads associated with these spurious features using human expertise. **Tip-adapter** (Zhang et al., 2021) is a robust, training-free method designed to enhance the accuracy of CLIP while preserving its zero-shot capabilities. We compare our method to both its training-free and training-based versions, even though our method does not require training at this stage. For the CelebA dataset, due to the unknown spurious features, we only compare our performance to that of Tip-adapter.

**Results** For both our method and Tip-Adapter, we evaluate performance using 10 samples per class. For TextSpan, we list its performance that used the entire test set to train textual descriptions, as it requires a lot of data. The performances for different methods and different CLIP pre-trained models on Binary Waterbirds are presented in Table 1. We report the average accuracy and the worst-case accuracy among the 4 birds groups (landbirds on land, landbirds on water, waterbirds on water and waterbirds on land). For CelebA, we show the performance of our method and Tip-Adapter in predicting the 'Young' or 'Old' attribute using CLIP-B/16, as detailed in Table 2. We highlight the following observations: (i) With the same number of samples per class, our method outperforms Tip-Adapter on both the Waterbirds and CelebA datasets. (ii) Despite utilizing fewer samples, incurring lower computational costs, and requiring no human intervention, our method generally outperforms TextSpan on the Waterbirds dataset. Furthermore, by incorporating a validation set to enhance our method, its performance can be further improved. Additional implementation details can be found in the Appendix A.2

**Stability Analysis** To evaluate the stability of our proposed identify-then-ablate method, we conduct a series of experiments on the Binary Waterbirds dataset. Specifically, we test with $n = 10, 20,$ and 30, where $n$ represents the number of randomly selected samples per class. For each sample size, we use four different random seeds for sample selection. Table 3 presents the mean and estimated standard deviation of the accuracies for each setting. Our method demonstrates stable editing capacity across different sample sizes and sample selections.

## 5.3 LOCALITY

In this section, we answer **Q3** by evaluating whether our two-phase approach, RefineCLIP, can achieve locality in a model editing scenario. We conduct tests on both a synthetic dataset and ImageNet-A.

**Datasets and Settings** ImageNet-R contains a diverse range of real-world images that CLIP can classify with ease. We select the 18 classes that have the highest number of cartoon-style images from ImageNet-R and combine them with the Binary Waterbirds dataset to create a new dataset consisting of 20 classes. Our goal with this dataset is to achieve editing success for the bird classes while preserving locality by minimizing side effects on the ImageNet-R classes, using only 10 samples from the Binary Waterbirds dataset per class.

Table 2: Average-group accuracy (%) and worst-group accuracy (%) for classifying 'Young' or 'Old' on the CelebA dataset using ViT-B/16.

| Method | Avg.↑ | Wst.↑ |
|---|---|---|
| Base | 70.1 | 43.8 |
| Tip-Adapter | 73.7 | 52.2 |
| Ours | **74.1** | **56.4** |

Similarly, we also evaluate RefineCLIP on ImageNet-A, a dataset consisting of real-world images misclassified by ResNet models. We select the 10 classes with the most images and focus on improving CLIP's performance on the 2 worst-performing classes, while minimizing any negative impact on the remaining eight. This is achieved using only 4 samples from each of the two target classes.

**Results** Unlike most model-editing methods that rely on prior knowledge, our proposed RefineCLIP is entirely data-driven. We compare it to standard fine-tuning using the same number of samples. As shown in Fig. 2, on the combined Waterbirds and ImageNet-R dataset, RefineCLIP achieves higher accuracy on the target classes than standard fine-tuning while significantly mitigating the performance degradation on other classes caused by overfitting—a critical issue with standard fine-tuning. On the ImageNet-A dataset, RefineCLIP strikes a balance between edit success and locality, achieving comparable accuracy to standard fine-tuning on the target classes while effectively reducing its side effects on unrelated classes.

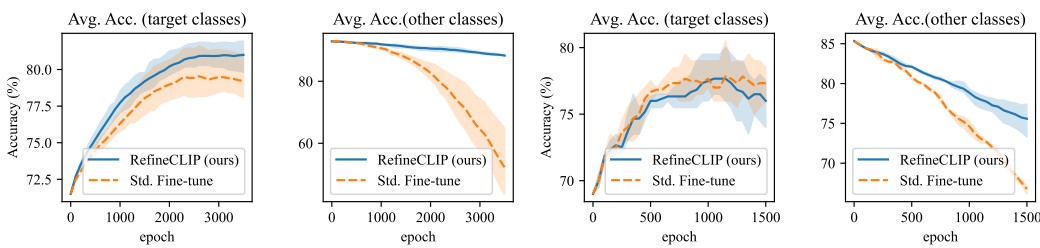

Figure 2: Performance Comparison of RefineCLIP and Standard Fine-Tuning with ViT-L/14. From left to right, the first two figures show the average accuracy for the target classes and other classes on the Waterbirds-ImageNet-R combined dataset, while the next two figures display the corresponding results for the ImageNet-A dataset. To reduce randomness, we use three random seeds for each method.

## 6 CONCLUSION

In this paper, we introduce a two-phase model editing framework for rectifying the prediction errors in CLIP which are caused by unknown spurious features. We show that the proposed measure effectively identifies the heads causing incorrect predictions and removing these identified features from the image representation repairs the model's performance. We further propose using representation adaption to refine CLIP features such that it reduces the influence of spurious features for incorrectly predicted data while preserving the prediction of unrelated data.

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

## A    EXPERIMENT DETAILS AND EXTRA RESULTS

### A.1    MISSING DETAILS IN SECTION 5.1

To make the known spurious cue, image background, the target cues for each comparing method, we select specific samples for each set. For methods (A) and (B), we use images where the background matches the bird type in the correctly predicted sets and mismatched backgrounds in the misclassified sets. For methods (C) and (D), to ensure the background serves as a spurious feature in both sets, we select only samples with mismatched backgrounds. We fix $K_1 = K_2 = 5$, resulting in a total of 30 samples, with different correctly predicted samples used across the method groups. We evaluate each attention head in the last four layers of the OpenCLIP ViT-L/14 model (Ilharco et al., 2021) and select the top $T = 15$ attention heads that cause the largest expected model shift for each method. The selected attention heads for each method are recorded in Table 4. The jointly selected attention heads and their corresponding TextSpan-generated textual descriptions are presented in Table 5.

Table 4: Attention heads identified by each method. Those jointly selected attention heads are highlighted in bold. (**L22**,H**0**) represents the (**0**+1)-th head in the (**22**+1)-th layer.

| Method (A) | Method (B) | Method (C) | Method (D) |
|---|---|---|---|
| **(L23, H2)** | **(L23, H2)** | **(L23, H2)** | **(L23, H2)** |
| **(L23, H5)** | **(L23, H5)** | **(L23, H6)** | **(L23, H6)** |
| (L22, H6) | (L22, H6) | **(L22, H1)** | **(L22, H1)** |
| (L22, H4) | (L22, H4) | **(L23, H5)** | **(L23, H5)** |
| (L23, H14) | (L23, H14) | **(L23, H8)** | **(L23, H8)** |
| **(L23, H3)** | **(L23, H3)** | (L23, H0) | (L23, H0) |
| (L22, H2) | (L22, H2) | (L22, H5) | (L22, H5) |
| (L21, H0) | (L21, H0) | **(L23, H3)** | **(L23, H3)** |
| **(L23, H12)** | **(L23, H12)** | **(L23, H9)** | **(L23, H9)** |
| **(L23, H8)** | **(L23, H8)** | (L23, H1) | (L23, H1) |
| (L22, H12) | (L22, H12) | (L21, H9) | (L21, H9) |
| **(L23, H9)** | **(L23, H9)** | (L22, H9) | (L22, H9) |
| **(L22, H1)** | **(L22, H1)** | **(L23, H12)** | **(L23, H12)** |
| **(L23, H6)** | **(L23, H6)** | (L20, H10) | (L20, H10) |
| (L21, H15) | (L21, H15) | (L22, H0) | (L22, H0) |

### A.2    MISSING DETAILS IN SECTION 5.2

We randomly select 10 samples per class (waterbirds and landbirds for the Waterbirds dataset, young and old celebrities for the CelebA dataset), with some correctly predicted by CLIP and others not. These samples are categorized into four groups based on their ground-truth labels and predicted labels. We then apply the comparison methods described in Section 3.2, skipping any methods that are infeasible due to insufficient data in the comparison set. If the comparison reference set lacks sufficient data, we assign zero as the average contribution for each attention head.

Next, we identify the attention heads contributing to incorrect predictions following the method presented in Section 3.2. Concretely, we obtain the top 15 attention heads, as a candidate list, using the proposed scores. Then we compute the utility defined in Eq. (10) for ablating the top $t$ heads in the candidate list. We repeat for all scores and perform the same ablation as the one with the largest utility. For our method enhanced with a validation set, we perform ablation on the top $t$ heads in each ordered candidate list and select the list that delivers the best performance on the validation set, rather than relying on the utility estimation.

Both our method and Tip-Adapter are evaluated in a zero-shot setting, leveraging CLIP's zero-shot capabilities without training additional classifiers. We show the results in Table 1 and Table 2.

Table 5: Common attention head with their top-5 results of TEXTSPAN.

| Layer 23, Head 5 | Layer 23, Head 3 |
|---|---|
| Intertwined tree branches | Bustling city square |
| Flowing water bodies | Serene park setting |
| A meadow | Warm and cozy indoor scene |
| A smoky plume | Modern airport terminal |
| Blossoming springtime blooms | Remote hilltop hut |
| **Layer 23, Head 8** | **Layer 23, Head 6** |
| Photograph with a red color palette | Picture taken in Sumatra |
| An image with cold green tones | Picture taken in Alberta, Canada |
| Timeless black and white | Picture taken in the geographical location of Spain |
| Image with a yellow color | Image taken in New England |
| Photograph with a blue color palette | Photo captured in the Arizona desert |
| **Layer 23, Head 2** | **Layer 23, Head 12** |
| Image showing prairie grouse | Image with polka dot patterns |
| Image with a penguin | Striped design |
| A magnolia | Checkered design |
| An image with dogs | Artwork with pointillism technique |
| An image with cats | Photo taken in Galapagos Islands |
| **Layer 23, Head 9** | **Layer 22, Head 1** |
| ornate cathedral | A semicircular arch |
| detailed reptile close-up | An isosceles triangle |
| Image with a seagull | An oval |
| A clover | Rectangular object |
| Futuristic space exploration | A sphere |

### A.3 COMPARE WITH CLIP-BASED PROMPT LEARNING APPROACHES IN FEW-SHOT SCENARIOS

Although RefineCLIP is primarily a model editing approach designed to achieve both edit success and edit locality, its data-driven identify-then-ablate editing component can also be used as a basis for comparison with various prompt learning approaches. Specifically, we compare this aspect of the model with recent CLIP-based prompt learning methods: CoOp Zhou et al. (2021), Plot Chen et al. (2023), and CLAP Cai et al. (2023). Since most of these methods are applied in a few-shot CLIP setting that involves training additional classifiers, rather than leveraging CLIP's zero-shot capabilities, we also train an additional classifier after applying our identify-then-ablate method. The experiments are conducted on the Waterbirds dataset. For each method, we randomly select 10 samples per class for training. As shown in Table 6, our method performs generally better than all of the other three baselines.

## B ABLATION STUDY AND SENSITIVITY ANALYSIS

**Spurious-feature-ablated model** To assess whether learning from the spurious-feature-ablated model developed in the first phase contributes to edit success, we conduct an ablation experiment. Specifically, we remove the component of the training objective related to this model and treat all available samples equally. In other words, we train the model to minimize the KL divergence with the initial model across all samples, instead of dividing them into two groups—one learning from the ablated model and the other from the initial model. The results in Fig. 4 highlight the significance of this component, as the model's capacity to edit the target class degrades significantly when this learning mechanism is excluded from the training process.

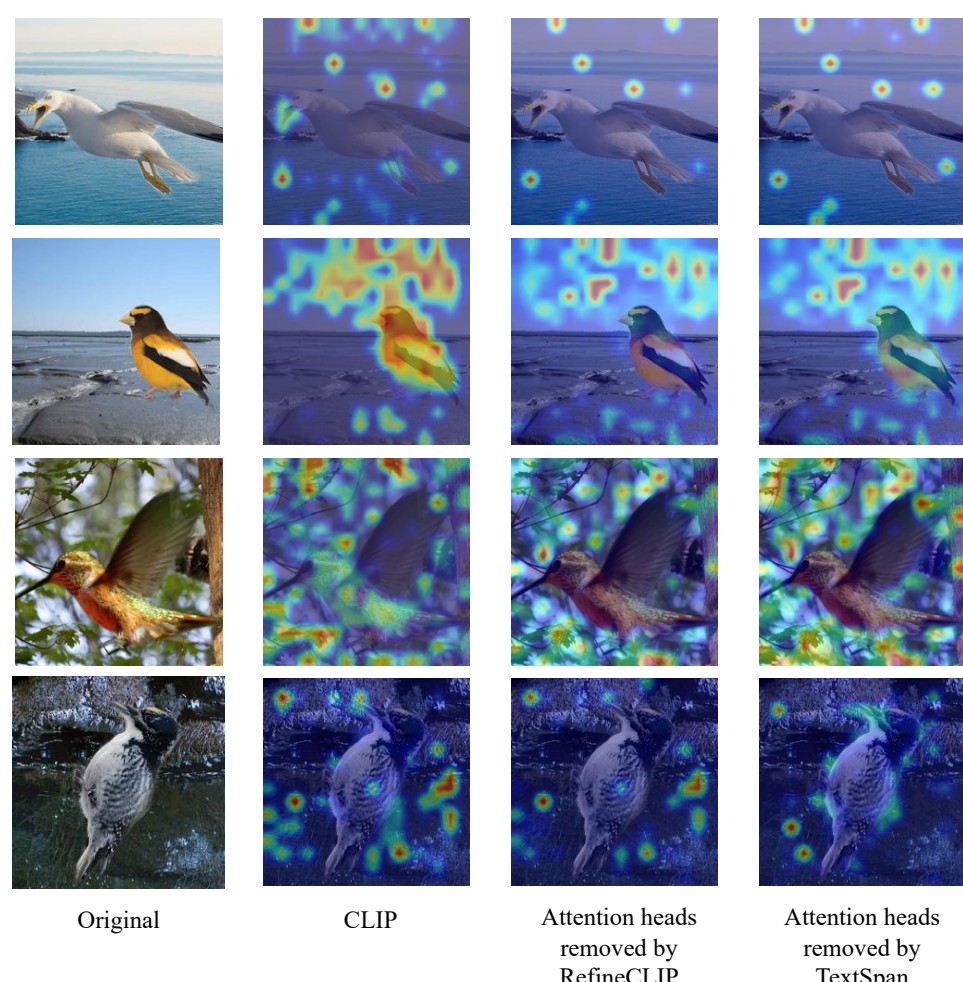

Original  CLIP  Attention heads removed by RefineCLIP  Attention heads removed by TextSpan

Figure 3: **Grad-Cam visualization.** We present four examples from the Waterbirds dataset, each illustrating the following from left to right: the original image, a heatmap showing the focus of the initial CLIP model, a heatmap highlighting the attention heads identified by RefineCLIP as spurious correlations based on all four scores, and a heatmap highlighting the attention heads selected by TextSpan as related to the background. In the task of classifying waterbirds and landbirds, domain knowledge identifies bird claws and beaks as causal features, while the background represents spurious correlations. As shown in the examples, the attention heads selected by RefineCLIP as spurious cues primarily focus on the images' backgrounds. Furthermore, compared to TextSpan, those attention heads selected by RefineCLIP demonstrates greater focus on spurious cues and less attention on causal features, despite TextSpan relying on prior domain knowledge and requiring significantly more human effort and computational resources.

**Ablation study of $\mathcal{L}_{\text{success}}(\theta)$ and $\mathcal{L}_{\text{locality}}(\theta)$** In accordance with Eq. (11), (12), and (13), the fine-tuning loss function is defined as a combination of the edit success loss, $\mathcal{L}_{\text{success}}(\theta)$, the edit locality loss, $\mathcal{L}_{\text{locality}}(\theta)$, and the cross-entropy loss, $\mathcal{L}_{\text{CE}}(\theta)$. To evaluate the contribution of the first two components to edit success and locality, we perform an ablation study on the 'Waterbirds + Imagenet-R' dataset by isolating each loss term. For clarity, the weight of the cross-entropy loss is fixed at 1, while the weights of $\mathcal{L}_{\text{success}}(\theta)$ and $\mathcal{L}_{\text{locality}}(\theta)$ are donated as $\alpha$ and $\beta$, respectively.

As illustrated in Fig. 5, the ablation study for $\mathcal{L}_{\text{locality}}(\theta)$ is performed by fixing the weights of $\mathcal{L}_{\text{CE}}(\theta)$ and $\mathcal{L}_{\text{success}}(\theta)$ at 1 and 0, respectively, while varying the weight $\beta$ of $\mathcal{L}_{\text{locality}}(\theta)$ from 0 to $10^8$. As $\beta$ increases, we observe a general improvement in the average accuracies of unseen

Table 6: Average-group accuracy (%) and worst-group accuracy (%) on the Waterbirds dataset using ViT-B/16. The results for each baseline are obtained using the public code released by the authors.

| Method | Avg.↑ | Wst.↑ |
|--------|-------|-------|
| Original CLIP | 80.4 | 72.3 |
| CoOp | **85.7** | 77.7 |
| Plot | 81.4 | 71.6 |
| CLAP | 81.9 | 72.3 |
| Ours | 85.2 | **81.0** |

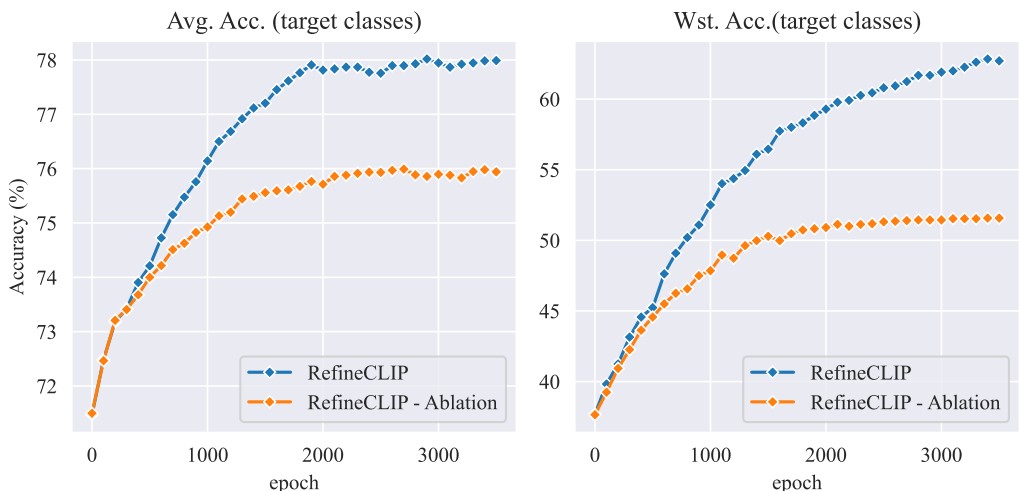

Figure 4: Performance comparison for the ablation study. (The weights of $\mathcal{L}_{\text{locality}}(\theta)$, $\mathcal{L}_{\text{locality}}(\theta)$, $\mathcal{L}_{\text{CE}}(\theta)$ are set to $10^6$, $10^4$, and 1, respectively.)

classes, accompanied by a decline in the average accuracies of target classes during the training. This behavior demonstrates the trade-off associated with the locality enhancement introduced by $\mathcal{L}_{\text{locality}}(\theta)$. Notably, when $\beta = 10^3$, the model achieves its highest peak average accuracy on the target classes during training. This finding suggests that a moderate value of $\beta$ can function as an effective regularizer, guiding the training process in a favorable direction.

Similarly, we conduct an ablation study for $\mathcal{L}_{\text{success}}(\theta)$ by varying its weight, $\alpha$, from 0 to $10^8$, while keeping the weights of $\mathcal{L}_{\text{CE}}(\theta)$ and $\mathcal{L}_{\text{locality}}(\theta)$ fixed at 1 and 0, respectively. As shown in Fig. 6, increasing $\alpha$ leads to a general improvement in the average accuracy of target classes during training, with the best performance achieved at $\alpha = 10^4$. Additionally, higher values of $\alpha$ also result in improved accuracies for other classes. This is because, although learning from the ablated model for edit success ($\mathcal{L}_{\text{success}}(\theta)$) counteracts the model's original locality objective ($\mathcal{L}_{\text{locality}}(\theta)$), it still helps mitigate over-fitting compared to standard fine-tuning.

Finally, comparing Fig. 6 with Fig. 5, we observe that when weighted equally, $\mathcal{L}_{\text{locality}}(\theta)$ demonstrates a stronger ability to preserve edit locality at the point where the average accuracy on the target classes peaks. For instance, when $\alpha = 10^4$ and $\beta = 0$, the average accuracy on other classes remains around 85% at the peak accuracy of the target classes. In contrast, when $\alpha = 0$ and $\beta = 10^4$, the average accuracy on other classes improves to nearly 90%. Conversely, $\alpha = 10^4$ and $\beta = 0$ achieve approximately 81% accuracy on the target classes, whereas $\alpha = 0$ and $\beta = 10^4$ achieve only about 77%. These observations suggest that $\mathcal{L}_{\text{locality}}(\theta)$ primarily emphasizes locality, while $\mathcal{L}_{\text{success}}(\theta)$ prioritizes edit success.

**Sensitivity analysis of $\alpha$ and $\beta$**

918
919
920
921
922
923
924
925
926
927
928
929
930
931
932
933
934
935
936
937
938
939

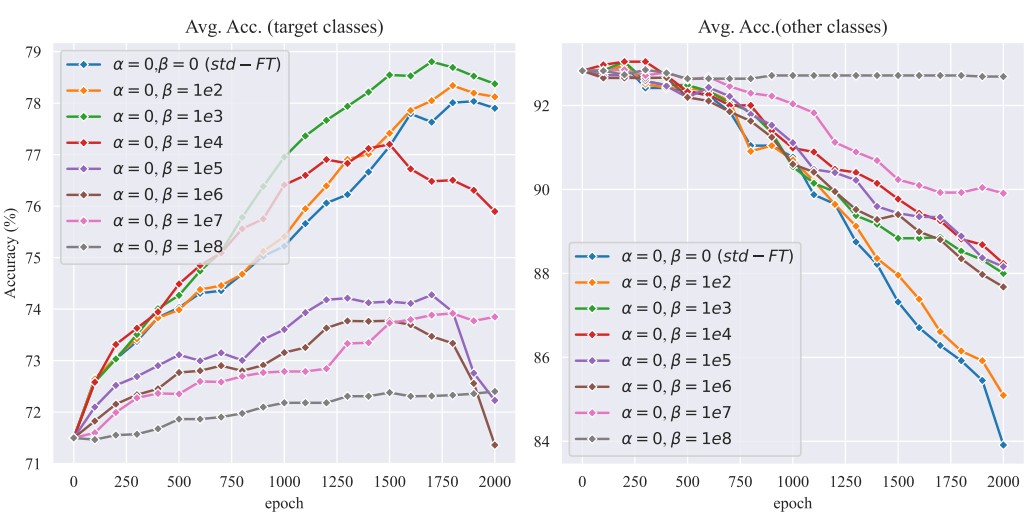

Figure 5: **Ablation study for $\mathcal{L}_{\textbf{locality}}(\theta)$.** STD-FT refers to Standard Fine-tuning.

940
941
942
943
944
945
946
947
948
949
950
951
952
953
954
955
956
957
958
959
960
961
962
963
964
965
966

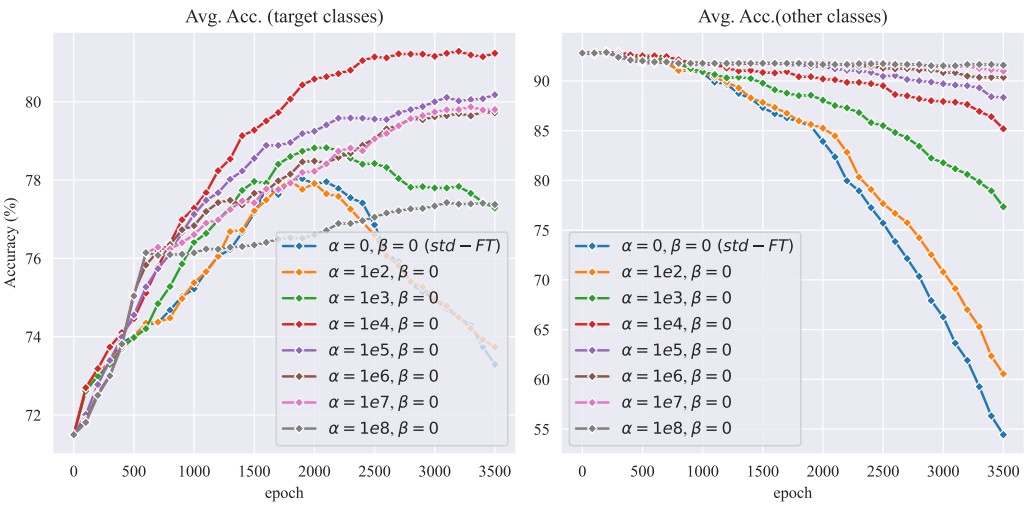

Figure 6: **Ablation study for $\mathcal{L}_{\textbf{success}}(\theta)$.** STD-FT refers to Standard Fine-tuning.

967
968
969
970
971

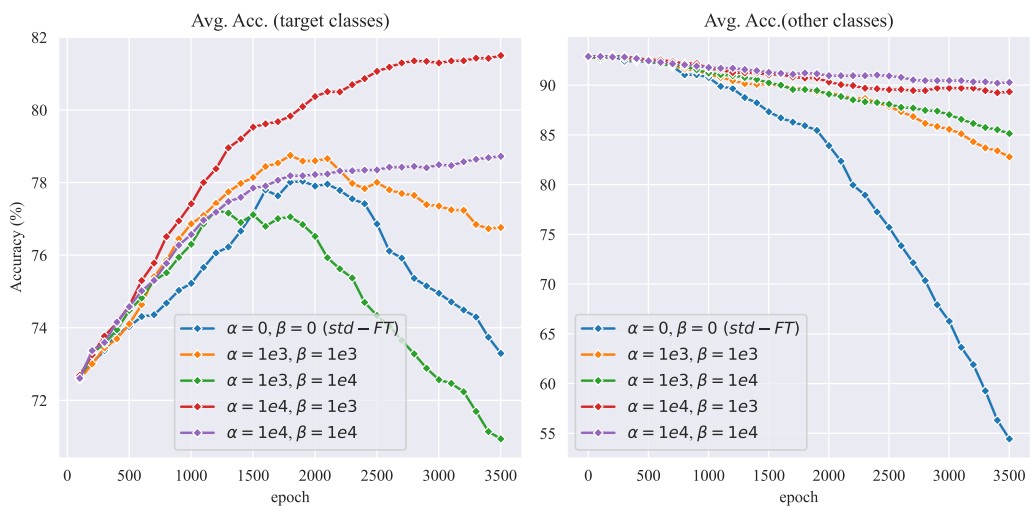

Figure 7: Sensitivity analysis on 'Waterbirds + ImageNet-R'

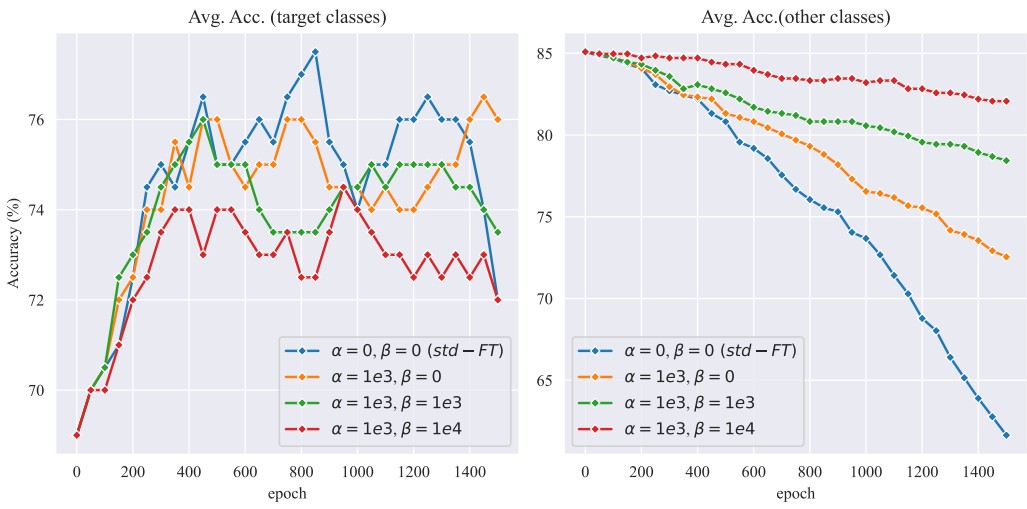

Figure 8: Sensitivity analysis on ImageNet-A

Based on the ablation study of $\mathcal{L}_{\text{success}}(\theta)$ and $\mathcal{L}_{\text{locality}}(\theta)$ on the 'Waterbirds+ImageNet-R' dataset, we observe that when the weights for $\mathcal{L}_{\text{success}}(\theta)$ and $\mathcal{L}_{\text{locality}}(\theta)$ are set between $10^3$ and $10^4$, our method achieves optimal performance in terms of both edit success and edit locality. To evaluate whether this weight range consistently delivers stable performance across different datasets, we conducted a sensitivity analysis of $\alpha$ and $\beta$ on 'Waterbirds+ImageNet-R' and 'ImageNet-A'. As shown in Figures 7 and 8, our method demonstrates stable edit success and locality within this weight range, outperforming standard fine-tuning. All the experiments are done based on CLIP-L/14.

**Qualitative Discussion on the Effectiveness of Individual Scores** In all the experiments conducted, when we select the final list of ablated attention heads based on their performance on the available samples, we observe that $\text{DE}_C^{l,h}$ and $\text{DE}_D^{l,h}$ generally outperform $\text{DE}_A^{l,h}$ and $\text{DE}_B^{l,h}$ in identifying better attention heads for ablation. In the CelebA dataset, ablating only the attention heads selected by $\text{DE}_A^{l,h}$ and $\text{DE}_B^{l,h}$ can even have negative effects, leading the model to predict all samples as belonging to a single class. This occurs because $\text{DE}_A^{l,h}$ and $\text{DE}_B^{l,h}$ are based on the assumption that the data can be fairly predicted without systemic bias. In cases where the model tends to predict everything

as a single class and relies on only a few features to distinguish between labels, as seen with the CelebA dataset, ablating those features (selected by $\text{DE}_A^{l,h}$ and $\text{DE}_B^{l,h}$ due to their negative impact on misclassified samples compared to correctly classified ones) can exacerbate the model's bias towards a single label.

## C  DERIVE THE IDENTIFICATION RESULTS FROM THE FOUR SCORES

In the first phase of RefineCLIP, we identify the most effective attention heads to ablate by following a systematic process:

1. **Score Calculation:** For each attention head in the final four layers, we calculate four scores: $\text{DE}_A$, $-\text{DE}_B$, $\text{DE}_C$, and $-\text{DE}_D$, using the available samples.

2. **Candidate List Generation:** For each score, we rank the attention heads in descending order based on their respective score and select the top $T$ heads to form a candidate list associated with that score.

3. **Generating Ablation Candidates:** For each candidate list, we iteratively ablate the top $t$ heads (from $t = 1$ to $T$). Let $\mathcal{S}_t$ represent the set of heads ablated at each step.

4. **Utility Evaluation:** For each ablation configuration $\mathcal{S}_t$, we evaluate the resulting model's performance by calculating the utility score (defined in Eq. 10) on the available samples. The set $\mathcal{S}_t$ that achieves the highest utility score is selected as the final list of attention heads to ablate.

To ensure clarity, we provide pseudo-code below outlining this strategy step by step.

---

**Algorithm 1** Derive the identification results from the four scores

---

**Input:** The breakdown of the image representation at the attention head level for each available sample.
**Output:** A set of attention heads selected for ablation.
**for** $\text{DE}_B, -\text{DE}_B, \text{DE}_C, -\text{DE}_D$ **do**
    Calculate the score for each attention head in the final four layers using the breakdown of image representations from the available samples.
    Order the attention heads in descending order based on their respective score and select the top $T$ heads as a candidate list associated with that score.
    **for** $t$ in [1,2,...,$T$] **do**
        Select the top $t$ attention heads from the candidate list to form $\mathcal{S}_t$.
        Ablate the attention heads from $\mathcal{S}_t$ in the breakdown of image representations from the available samples.
        Calculate the utility score based on the ablated image representations from the available samples, represented as $\text{U}(\mathcal{S}_t)$
    **end for**
**end for**
Compare all available $\text{U}(\mathcal{S}_t)$ values and select the $\mathcal{S}_t$ with the largest $\text{U}(\mathcal{S}_t)$ as $\mathcal{S}_t^*$.
**return** $\mathcal{S}_t^*$

---

**Algorithm 2** Refine CLIP

---

Obtain the image representation $\boldsymbol{E}_{\text{image}}(x)$ and the breakdown of the image representation at the attention head level for each available sample.
**/* Phase 1 */**
Call Algorithm 1 to obtain a set of attention heads selected for ablation as $\mathcal{S}_t^*$
Compute the image representation $\boldsymbol{E}_{\text{image}}^{\text{ablated}}(\boldsymbol{x}, \mathcal{S})$ after removing the influence of these spurious features by Eq. (5) using $\mathcal{S} = \mathcal{S}_t^*$
**/* Phase 2 */**
Obtain $\text{diag}(\theta)$ by minimize the objective function in Eq. (13)

---

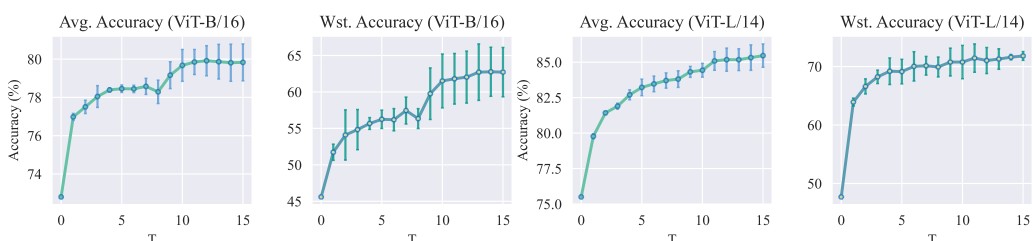

Figure 9: Sensitivity analysis for $T$

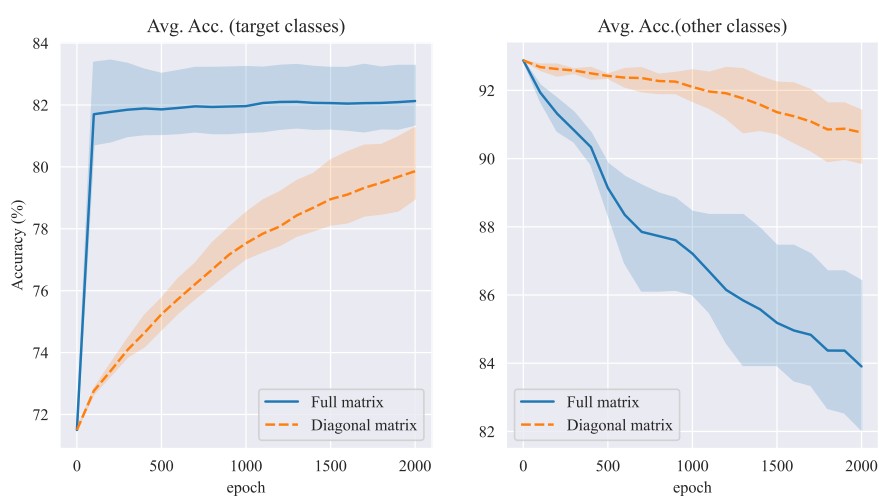

Figure 10: Full matrix vs diagonal matrix on 'Waterbird+ImageNet-R'

As described above, the hyper-parameter $T$ does not directly define the number of attention heads to be ablated; rather, it sets the size of the pool from which we select attention heads based on their utility scores. As a result, the improvement gained from ablation does not change significantly once $T$ becomes large enough. To support this claim, we conducted a sensitivity analysis on the Waterbirds dataset using both ViT-B/16 and ViT-L/14. We used three random seeds to minimize variability. As shown in Fig. 9, when $T$ exceeds 10, the performance improvements become stable.

## D  UPDATE STRATEGIES

In the second phase, RefineCLIP's primary contribution is the design of a loss function that effectively balances edit success with edit locality. Regarding the update strategy, while our approach supports various strategies—including a trainable full matrix—we primarily adopt the diagonal matrix update strategy introduced in Section 3.3.

This choice is motivated by our method's focus on scenarios where data for correcting errors is scarce, and no additional data is available for unrelated classes. Although a full matrix introduces more trainable parameters, which can improve training on target classes, it increases the risk of overfitting to the limited available data, ultimately compromising edit locality.

To investigate this trade-off, we conducted experiments on the 'Waterbird+ImageNet-R' and 'ImageNet-A' datasets, comparing the performance of diagonal and full matrix update strategies. The experimental setup matches that described in Section 5.3, and we used three random seeds to mitigate randomness. The results are presented in Fig. 10 and Fig. 11.

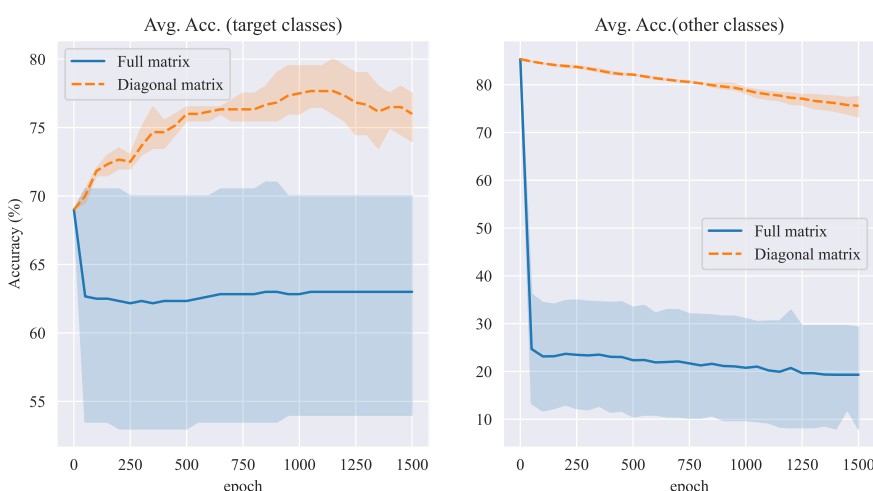

Figure 11: Full matrix vs diagonal matrix on ImageNet-A

As shown in Fig. 10, on the 'Waterbird+ImageNet-R' dataset, the full matrix strategy achieved better edit success and faster convergence, albeit with a slight loss in locality, likely due to the availability of 10 samples per class for training. However, on the 'ImageNet-A' dataset (Fig. 11), where only 4 samples per class were available, the full matrix strategy failed completely.

LoRAHu et al. (2021), typically used for fine-tuning parameters within transformers, can also be integrated with RefineCLIP to fine-tune the projection matrix. The rank serves as a hyperparameter that balances fine-tuning capacity and potential overfitting by controlling the scale of trainable parameters. We conducted experiments on the 'ImageNet-A' dataset, where the full matrix strategy previously failed, to evaluate the impact of rank on performance. For each rank, we used three random seeds to mitigate randomness. The results are presented in Fig. 12.

In summary, RefineCLIP supports various update strategies, including full matrix, diagonal matrix, and LoRA. The scale of trainable parameters represents a trade-off between fine-tuning capacity and the risk of over-fitting, which can ultimately compromise edit locality. We adopt the diagonal matrix as the standard and representative update strategy in Section 3.3 because it demonstrates stability even in tasks with extremely scarce data while largely preserving edit locality.

## E  PROPERTY OF THE FOUR DIRECT EFFECTS SCORES

In the framework of CLIP, the cosine similarity function is commonly employed to assess the similarity between the representations of images and texts. This function normalizes the magnitude of the vectors and only considers their direction, which is crucial for comparing vectors of different scales. However, if we use the dot product similarity, we observe interesting properties: $\mathrm{DE}_A^{l,h} = -\mathrm{DE}_B^{l,h}$ and $\mathrm{DE}_C^{l,h} = -\mathrm{DE}_D^{l,h}$. We formally present this property in Proposition 1 with proof. Transitioning back to cosine similarity, which is a normalized form of the dot product, the relationships approximately hold: $\mathrm{DE}_A^{l,h} \approx -\mathrm{DE}_B^{l,h}$ and $\mathrm{DE}_C^{l,h} \approx -\mathrm{DE}_D^{l,h}$. This insight is helpful for understanding the relation between the proposed scores.

**Proposition 1.** *By using the inner product as a similarity function, we have $DE_A^{l,h} = -DE_B^{l,h}$ and $DE_C^{l,h} = -DE_D^{l,h}$.*

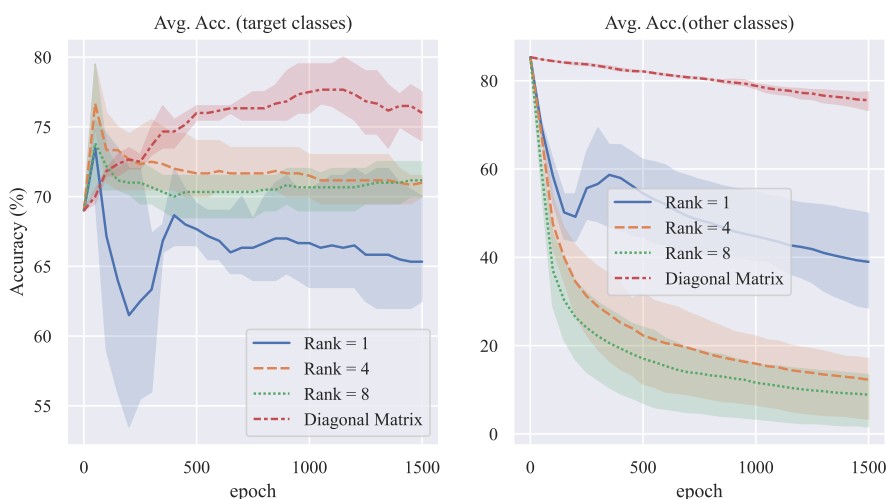

Figure 12: LoRA vs diagonal matrix on ImageNet-A

*Proof.* Combining Eq. (6), Eq. (7) and Eq. (8), ignoring the differences in normalization when calculating cosine similarity, we can obtain:

$$\Delta^{l,h}(\boldsymbol{x}, y, \boldsymbol{h}) = \text{sim}\Big(\boldsymbol{E}_{\text{image}}(\boldsymbol{x}) - \boldsymbol{P}\boldsymbol{h}^{l,h}(\boldsymbol{x}) + \boldsymbol{P}\boldsymbol{h}, \boldsymbol{E}_{\text{text}}(y)\Big) - \text{sim}\Big(\boldsymbol{E}_{\text{image}}(\boldsymbol{x}), \boldsymbol{E}_{\text{text}}(y)\Big)$$

$$= \Big(\boldsymbol{E}_{\text{image}}(\boldsymbol{x}) - \boldsymbol{P}\boldsymbol{h}^{l,h}(\boldsymbol{x}) + \boldsymbol{P}\boldsymbol{h}\Big)^{\top} \boldsymbol{E}_{\text{text}}(y) - \boldsymbol{E}_{\text{image}}(\boldsymbol{x})^{\top} \boldsymbol{E}_{\text{text}}(y)$$

$$= \big(\boldsymbol{P}\boldsymbol{h} - \boldsymbol{P}\boldsymbol{h}^{l,h}(\boldsymbol{x})\big)^{\top} \boldsymbol{E}_{\text{text}}(y)$$

Plugging it into $\text{DE}_A^{l,h}$ and $\text{DE}_B^{l,h}$, we have

$$\text{DE}_A^{l,h} = \mathbb{E}_{\boldsymbol{x}_i \in \mathcal{W}_{y_e, \hat{y}_e}} \Big[ \Delta^{l,h}\Big(\boldsymbol{x}_i, y_e, \bar{\boldsymbol{h}}_{\mathcal{C}_{y_e}}^{l,h}\Big) - \Delta^{l,h}\Big(\boldsymbol{x}_i, \hat{y}_e, \bar{\boldsymbol{h}}_{\mathcal{C}_{y_e}}^{l,h}\Big) \Big]$$

$$= \mathbb{E}_{\boldsymbol{x}_i \in \mathcal{W}_{y_e, \hat{y}_e}} \Big[ \Big(\boldsymbol{P}\bar{\boldsymbol{h}}_{\mathcal{C}_{y_e}}^{l,h} - \boldsymbol{P}\boldsymbol{h}^{l,h}(\boldsymbol{x}_i)\Big)^{\top} \boldsymbol{E}_{\text{text}}(y_e) - \Big(\boldsymbol{P}\bar{\boldsymbol{h}}_{\mathcal{C}_{y_e}}^{l,h} - \boldsymbol{P}\boldsymbol{h}^{l,h}(\boldsymbol{x}_i)\Big)^{\top} \boldsymbol{E}_{\text{text}}(\hat{y}_e) \Big]$$

$$= \Big(\boldsymbol{P}\bar{\boldsymbol{h}}_{\mathcal{C}_{y_e}}^{l,h} - \mathbb{E}_{\boldsymbol{x}_i \in \mathcal{W}_{y_e, \hat{y}_e}} \boldsymbol{P}\boldsymbol{h}^{l,h}(\boldsymbol{x}_i)\Big)^{\top} \boldsymbol{E}_{\text{text}}(y_e) - \Big(\boldsymbol{P}\bar{\boldsymbol{h}}_{\mathcal{C}_{y_e}}^{l,h} - \mathbb{E}_{\boldsymbol{x}_i \in \mathcal{W}_{y_e, \hat{y}_e}} \boldsymbol{P}\boldsymbol{h}^{l,h}(\boldsymbol{x}_i)\Big)^{\top} \boldsymbol{E}_{\text{text}}(\hat{y}_e)$$

$$= \Big(\mathbb{E}_{\boldsymbol{x}_i \in \mathcal{C}_{y_e}} \boldsymbol{P}\boldsymbol{h}^{l,h}(\boldsymbol{x}_i) - \boldsymbol{P}\bar{\boldsymbol{h}}_{\mathcal{W}_{y_e, \hat{y}_e}}^{l,h}\Big)^{\top} \boldsymbol{E}_{\text{text}}(y_e) - \Big(\mathbb{E}_{\boldsymbol{x}_i \in \mathcal{C}_{y_e}} \boldsymbol{P}\boldsymbol{h}^{l,h}(\boldsymbol{x}_i) - \boldsymbol{P}\bar{\boldsymbol{h}}_{\mathcal{W}_{y_e, \hat{y}_e}}^{l,h}\Big)^{\top} \boldsymbol{E}_{\text{text}}(\hat{y}_e)$$

$$= \mathbb{E}_{\boldsymbol{x}_i \in \mathcal{C}_{y_e}} \Big[ \Big(\boldsymbol{P}\boldsymbol{h}^{l,h}(\boldsymbol{x}_i) - \boldsymbol{P}\bar{\boldsymbol{h}}_{\mathcal{W}_{y_e, \hat{y}_e}}^{l,h}\Big)^{\top} \boldsymbol{E}_{\text{text}}(y_e) - \Big(\boldsymbol{P}\boldsymbol{h}^{l,h}(\boldsymbol{x}_i) - \boldsymbol{P}\bar{\boldsymbol{h}}_{\mathcal{W}_{y_e, \hat{y}_e}}^{l,h}\Big)^{\top} \boldsymbol{E}_{\text{text}}(\hat{y}_e) \Big]$$

$$= -\text{DE}_B^{l,h}$$

Similar inference holds for $\text{DE}_C^{l,h}$ and $\text{DE}_D^{l,h}$.

$\square$

## F  LIMITATIONS AND DIRECTIONS FOR FUTURE RESEARCH

In this work, we focus on identifying the model components related to spurious correlations by analyzing the direct effects of attention heads within the image encoder of CLIP-ViT. Specifically, we examine how these attention heads impact the image representation. However, this analysis does not account for the indirect effects of attention heads, which could also influence model behavior in subtle ways. The exclusion of these indirect effects represents a key limitation of our current approach. Future work could explore methods to incorporate or approximate the indirect relationships between attention heads, potentially improving the robustness of spurious correlation identification.

