# OpenReview forum: "Model Editing for CLIP with Unknown Spurious Correlations in Visual Encoder"
_ICLR.cc/2025/Conference — Submitted to ICLR 2025_

### Official Review · Reviewer_9oH7 · 2024-10-31

**Soundness:** 3
**Presentation:** 2
**Contribution:** 2
**Rating:** 5
**Confidence:** 4

**Summary:**

This paper addresses the issue of CLIP capturing spurious correlations during feature extraction, leading to prediction errors, and proposes a model editing method for CLIP called RefineCLIP. Specifically, RefineCLIP ranks each attention head in every layer based on its contribution to correct and incorrect classifications and sequentially ablates the outputs of certain attention heads. The contribution is determined by the change in the model's features and the class text features before and after ablation. Additionally, the model introduces an extra diagonal matrix as model parameters and fine-tunes it. The paper conducts experiment on the Waterbirds Dataset to validate the performance of the proposed method.

**Strengths:**

1. The method for evaluating the contribution of each attention head is novel.

**Weaknesses:**

1. Some existing studies [1] indicate that model editing can harm the generalization ability of the base model, contrary to the claim in this paper that it does not affect unrelated data. The experimental results in Figure 2 also indicate this.
2. The proposed method requires training and cannot be applied to unlabeled images.
3. The readability of the paper is not good. Including a schematic diagram of the proposed method could improve this.
4. The comparison methods are insufficient. This paper only compares with TextSpan [2] and the training-free methods Tip-adapter [3], without comparing with more debiasing methods for CLIP.
5. There is a lack of visualization results. From the visualization in Figure 1, it is still unclear how the attention regions of the model changed during classification before and after ablation.
6. The ablation experiments are insufficient. The roles of the three losses mentioned in Section 3.3 are not analyzed. The weights of these three loss functions are also not clearly defined.

[1] Gu J C, Xu H X, Ma J Y, et al. Model editing harms general abilities of large language models: Regularization to the rescue[J]. arXiv preprint arXiv:2401.04700, 2024.
[2] Gandelsman Y, Efros A A, Steinhardt J. Interpreting CLIP's Image Representation via Text-Based Decomposition[J]. arXiv preprint arXiv:2310.05916, 2023.
[3] Zhang R, Fang R, Zhang W, et al. Tip-adapter: Training-free clip-adapter for better vision-language modeling[J]. arXiv preprint arXiv:2111.03930, 2021.

**Questions:**

1. In the ablation experiment shown in Figure 3, the results are explained as coming from learning from the ablated model and the initial model. However, when all elements of the diagonal matrix are 1, the loss function in Equation 11 and 12 seems to be 0, making these loss functions meaningless. What are the details of this ablation experiment? How is the diagonal matrix used for fine-tuning initialized?

---

> ### Author Response · Authors · 2024-11-23
> **Response to Reviewer 9oH7 (Part 1/2)**
>
> We thank the reviewer for providing valuable feedback. Any modifications made to the paper are highlighted in blue. Please kindly let us know whether you have any further concerns.
>
> **Q1.1**: How is the diagonal matrix? When all elements of the diagonal matrix are 1, the loss function in Equation 11 and 12 seems to be 0, making these loss functions meaningless
>
> > - The diagonal matrix is initialized with all diagonal elements set to 1. This setup ensures that the model's initial predictions match those of the original model. Importantly, because these predictions differ from those of the spurious feature-removed model obtained in phase 1, **the loss in Eq. (11) is not zero at the start**.
> > - **It is a misunderstanding to consider a loss term meaningless if it is initially zero.** Our objective function, as shown in Eq. (13), includes **three loss terms. Even if the loss term in Eq. (12) begins at zero, it can become positive as the model updates to minimize the other two loss terms.** Throughout the optimization process, this loss term helps the model maintain the output of correctly predicted samples. Simultaneously, the loss term in Eq. (11) guides the model to align its predictions with those of the spurious feature-removed model for incorrectly predicted samples. The optimal model is achieved when the weighted sum of these three loss terms is minimized.
>
> **Q1.2**: What are the details of this ablation experiment shown in Fig.3?
>
> > - The ablation experiment (in Lines 803-809) aims at evaluating the impact of leveraging the ablated model from phase 1 on edit success in phase 2.
> > - **To test this, we exclude the loss term in Eq. (11) from the objective function in Eq. (13)**, and verify that the performance decreases. Without Eq. (11), no information from ablated model is provided for misclassified samples. To compensate for the removal of Eq. (11), we expand the loss in Eq. (12) to include all samples, not just the correctly predicted ones.
> > - By conducting this ablation experiment, we demonstrate the importance of the ablated model in enhancing the model's ability to edit successfully, thereby validating our approach.
>
>
> **W1**: Some existing studies [1] indicate that model editing can harm the generalization ability of the base model, contrary to the claim in this paper that it does not affect unrelated data.
>
> > We would like to clarify that our work **does not contradict** the findings presented in [1].
> > - Firstly, as outlined in [1], the three primary goals in model editing are reliability, generalization, and locality. In our work, as mentioned in Lines 52-53, we follow [2] to combine the first two goals under the term "editing success." We understand that your concern pertains to the potential negative impact on the "locality" of the base model.
> > - Secondly, "editing success" and "locality" are standard metrics for evaluating model editing methods. These metrics are typically assessed through accuracy on data related to the corrected samples (inside the editing scope) and unrelated data (outside the editing scope), respectively. While achieving 100% accuracy for both metrics is the ultimate goal, no current methods can guarantee perfect performance on both. Our claim that our method achieves a certain level of edit locality does not imply that it has zero impact on unrelated data.
> > We appreciate your feedback and hope this clarifies our position.
> >
> > [1] Y. Yao, P. Wang, B. Tian, et al., Editing large language models: Problems, methods, and opportunities. 2023.
> > [2] Eric Mitchell, Charles Lin, Antoine Bosselut, Chelsea Finn, and Christopher D Manning. Fast model editing at scale. ICLR 2022.

---

> ### Author Response · Authors · 2024-11-23
> **Response to reviewer 9oH7 (Part 2/2)**
>
> **W2.1: The proposed method requires training**
>
> > - We would like to highlight that our method also includes a **training-free** version, which is detailed as the ablated model in Phase 1. This version can correct wrongly-predicted samples, although it does not guarantee editing locality.
> > - For tasks where editing locality is not a priority, such as binary classification tasks on the Waterbirds and CelebA datasets (refer to Section 5.1), the ablated models from Phase 1 perform effectively. This phase is training-free and requires minimal computational resources, making it efficient to run even on a CPU.
> > - In scenarios where editing locality is essential, the model from Phase 2 is more appropriate. While Phase 2 does involve training, the computational cost is relatively low and stable, even for large-scale models. This is because we only optimize the diagonal matrix, with the number of trainable parameters being equivalent to the feature dimension, typically 768 for CLIP-ViT.
> > - Additionally, the features are not optimized during training and can be precomputed with a single forward pass through the model, resulting in negligible training cost.
> >
> > To provide a clearer understanding, we present the training time for **100 epochs** on an A100 GPU for each version of CLIP:
> >
> |      | CLIP-ViT-B/16 | CLIP-ViT-L/14 | CLIP-ViT-H/14 |
> |------|---------------|---------------|---------------|
> | Time | 125 seconds   | 136 seconds   | 207 seconds   |
>
>
> **W2.2**: The proposed method cannot be applied to unlabeled images
>
> > Our method focuses on model editing to correct specific errors, which inherently requires some labeled samples to identify these errors. As demonstrated in our stability analysis (Page 9 Table 3), only a small number of labeled samples are necessary to achieve significant improvements.
> >
> > Moreover, model editing fundamentally relies on having data with both current predictions and post-editing predictions. Therefore, applying our method to completely unlabeled data is not feasible, as it contradicts the essential requirements of model editing.
>
> **W3**: The readability of the paper is not good. Including a schematic diagram of the proposed method could improve this.
>
> > Thank you for your valuable suggestion. We have followed your advice to include a schematic diagram of the proposed method in Page 2 Fig 1 to enhance readability. Additionally, we have provided a detailed algorithm in Algorithms 1 and 2 (Appendix: Page 20) to further illustrate the proposed methods.
>
>
> **W4**: Comparison with debiasing methods for CLIP.
>
> > - As mentioned in Lines 69-71, traditional debiasing methods often rely on prior knowledge of the specific spurious correlations causing bias. In contrast, our approach is entirely data-driven and does not require any domain-specific knowledge. Therefore, comparing our method directly with these debiasing methods may **not be entirely fair**, as they leverage additional information that our method does not.
>
> > - To the reviewer's concern, we have conducted additional experiments to compare our method with three recent debiasing methods: CoOp, CLAP, and PLOT, as suggested by Reviewer AZn1. The detailed information is provided in Appendix:Lines 791–800 and Page 17 Table 6. As summarized below, our method performs generally better than all of the other three basedlines on Waterbirds.
>
> |              | Avg. Acc | Wst. Acc |
> |--------------|----------|----------|
> | original CLIP| 80.4%   | 72.3%   |
> | CoOp| **85.7%**   | $\underline{77.7}$%   |
> | Plot| 81.4%   | 71.6%   |
> | CLAP| 81.9%   | 72.3%   |
> | **Ours**| $\underline{85.2}$ %  | **81.0%**  |
>
> **W5: Lack of Visualization Results**
>
> > Thank you for your suggestion. We have added additional Grad-CAM visualizations with more detailed explanations in Page 16 Fig 3. Please kindly refer to the modified version of the paper.
>
>
> **W6: Missing Sensitivity and Ablation Analysis of Loss Weights.**
>
> >Thank you for your advice. We have follow your suggenstion to add ablation analysis and sensitivity analysis experiments of the >loss weights $(\alpha, \beta)$. The results of these comparisons are presented in Appendix: Lines 856–1018 and Figures 5 through >8. In summary, the ablation study demonstrates that $L_{\text{locality}}(\theta)$  primarily emphasizes locality, while $L_{success}>(\theta)$ focuses on achieving edit success. Additionally, the sensitivity analysis reveals that when the weights fall between >$10^3$ and $10^4$, our method achieves optimal performance in terms of both edit success and locality across the >'Waterbirds+ImageNet-R' and 'ImageNet-A' datasets.

---

> ### Author Response · Authors · 2024-11-23
> **Summary of rebuttal revisions**
>
> We sincerely thank the reviewers for their thoughtful and constructive feedback. Based on the comments, we have made several revisions to improve the clarity, robustness, and comprehensiveness of the manuscript. Below is a summary of the revisions made in response to the reviewers’ suggestions.
>
> **Additional experiments and anlysis**
> >1. Comparison Between Tip-Adapter (Training Version) and Our Training-Free Stage 1 on the Waterbirds Dataset. (Page 9 Table 1)
> >2. More experiments for Our Stage 2 (Page 10 Fig 2)
> >3. Comparison with CLIP-Based Prompt learning approaches (CoOp, CLAP, PLOT) in few-shot scenarios. (Appendix: Page 15 & Table 6)
> >4. Ablation study of $L_{success}(\theta)$ and $L_{locality}(\theta)$ (Appendix: Page 16-17 & Fig 5,Fig 6)
> >5. Sensitivity study of $L_{success}(\theta)$ and $L_{locality}(\theta)$ (Appendix: Page 18-19 & Fig 7,Fig 8)
> >6. Sensitivity analysis for $T$ (Appendix:Page 21 Fig 9)
> >7. Analysis of different update strategy. (Appendix:Page 21 Fig 10, Fig 11, Fig 12)
>
> **Additional visualizations**
> >1. An overview of the proposed method in Page 2. Fig.1.
> >2. More heatmap visualization using Grad-CAM (Appendix: Page 16 Fig 3)
>
> **Other revisions**
> >1. Add callback sentences at the end of Section 3.2 to ensure logical coherence and smooth transition to the subsequent sections.
> >2. More details about how to derive the identification results from the four scores. (Appendix: Page 20 & Algorithm 1)
> >3. An overview of the proposed method  (Appendix: Page 20  Algorithm 2)
> >4. Limitations and directions for future research (Appendix: Page 23)

---

> ### Comment · Reviewer_9oH7 · 2024-11-24
>
> The adjustments made by the authors result in improvements in several aspects of this paper, including experiments and readability. In light of these positive changes, we are inclined to increase the rating.

---

> ### Author Response · Authors · 2024-11-26
>
> Dear reviewer 9oH7, Thank you for your reply. We are delighted to know that our responses have effectively addressed your concerns, leading to an increase in your rating. If you have any additional suggestions or concerns, please do not hesitate to share them with us.

---

### Official Review · Reviewer_yUDY · 2024-11-02

**Soundness:** 3
**Presentation:** 3
**Contribution:** 2
**Rating:** 5
**Confidence:** 3

**Summary:**

The paper introduces a novel, data-driven method for editing CLIP models. The proposed approach comprises two main phases:

1. **Detection and Ranking of Faulty MSA Heads**: This phase involves calculating DE scores to evaluate changes in similarity to text features. This is achieved by replacing the Multi-Head Self-Attention (MSA) head features with averaged features from either correctly or incorrectly classified labels. The DE scores enable the ranking of MSA heads, thereby identifying which heads are faulty and may require editing.
2. **Training the Projection Matrix for Localized Edits**: To facilitate successful and localized edits, the method involves training a projection matrix, denoted as $\theta$, using a specifically designed loss function. This loss function ensures that the edits made to the MSA heads do not compromise the model's overall performance and maintain the desired locality of the changes.

**Strengths:**

The paper is clearly structured and easy to understand. The experimental results appear to be good.

**Weaknesses:**

1. **Comparison with Trainable Versions of Existing Methods**: The paper fails to consider the trainable version of Tip-Adapter. Since the proposed method involves training, it would be reasonable to finetune Tip-Adapter for a fairer evaluation.
2. **Insufficient Emphasis on Contributions**: The presentation does not adequately highlight the unique contributions of the proposed method. The authors note that the first stage resembles Gandelsman et al. and the second stage aligns with Santurkar et al., yet claim that the method is data-driven and does not require prior knowledge. However, previous works like Tip-Adapter also fall into this category, which undermines the novelty of the proposed approach.
3. **Inconsistent Experimental Design**: It is puzzling that the authors conduct experiments on the Waterbirds dataset for edit success, yet switch to different datasets (CelebA and ImageNet-R/A) for edit locality. It would be better to explain the motive of such a design.
4. **Limited Experimental Scope**: The experiments would benefit from a broader scope, particularly regarding success edits across multiple datasets. In contrast, Tip-Adapter has been evaluated on 10 different datasets, which provides a more comprehensive understanding of performance. Expanding the testing to include additional datasets would enhance the credibility of the findings.

**Questions:**

Besides my concerns raised in the Weakness, I have a few more questions.
1. **Visualization of Method Comparisons**: Could the authors provide an alternative version of Fig. 1 that displays heatmaps for each method separately, potentially including more images? This would enhance the understanding of the proposed method's superiority in detecting faulty heads.
2. **Impact of MSA Heads on Performance**: Does the number of Multi-Head Self-Attention (MSA) heads, denoted as TTT, influence the performance of the proposed method? A discussion and analysis regarding this aspect would be beneficial.

---

> ### Author Response · Authors · 2024-11-23
> **Response to reviewer yUDY ( Part1/2)**
>
> Thank you sincerely for your thoughtful feedback on our work. We are particularly grateful for your recognition of the various aspects of our research. Below, we have provided a detailed explanation for your remaining concern as follows.
>
> **Q1**： Can provide more heatmaps images in Fig.1?
> > We have included more heatmap images for our method and other methods in Fig.3 (in Page 16) with more detailed description.
> >
> > To summarize,  In the task of classifying waterbirds and landbirds, domain knowledge identifies bird claws and beaks as causal features, while the background represents spurious correlations. As shown in the examples, the attention heads selected by RefineCLIP as spurious cues primarily focus on the images’ backgrounds. Furthermore, compared to TextSpan, those attention heads selected by RefineCLIP demonstrates greater focus on spurious cues and less attention on causal features, despite TextSpan relying on prior domain knowledge and requiring significantly more human effort, computational resources.
>
>
> **Q2. Impact of MSA Heads on Performance.**
> > Thank you for your insightful comment. We appreciate the opportunity to clarify this aspect.
> >
> > - The hyperparameter $T$, as mentioned in Line 271, refers to the size of the candidate pool of attention heads, not the final number of MSA heads used in our method. From this pool, we employ a utility score to determine the optimal subset of heads for ablation.
> >
> > To address your concern, we have included a more detailed explanation in Appendix Page 20 to enhance clarity on this matter.
> >
> > Additionally, we have conducted additional sensitivity analysis and included new discussion on how the choice of $T$ impact performance, which can be found in Appendix Page 21, Fig.9. It shows that the improvement gained from ablation does not change significantly once $T$ becomes large enough.
>
>
> **Weakness 1. Comparison with trainable versions of Tip-Adatpor**
>
> >We would like to clarify that we are comparing Tip-Adapter in our training-free Stage 1, so it is appropriate to consider only the training-free version of Tip-Adapter. In Stage 2, which involves fine-tuning, our method has distinct goals from Tip-Adapter. Specifically, we aim to achieve both edit success and locality, as we are focused on model editing. On the other hand, Tip-Adapter does not appear to address the challenge of unseen classes during training.
>
> >Nevertheless, we have included the comparison results with the trainable version of Tip-Adapter in Page 9, Table 1, and we also present the results (ViT-B/16,ViT-L/14,ViT-H/14) here for clarity.
>
> |                   | Avg  | Wst  | Avg  | Wst  | Avg  | Wst  |
> |-------------------|------|------|------|------|------|------|
> | Base              | 72.8 | 45.6 | 75.5 | 47.7 | 68.6 | 37.2 |
> | Tip-Adapter       | 74.4 | 46.9 | 77.4 | 52.6 | 70.3 | 38.0 |
> | Tip-Adapter-train | 72.3 | 49.9 | 78.0 | 52.2 | 74.8 | $\textbf{59.3}$ |
> | Ours              | $\textbf{81.1}$ | $\textbf{61.4}$ | $\textbf{85.5}$ | $\textbf{72.1}$ | $\textbf{75.9}$ | 51.3 |

---

> ### Author Response · Authors · 2024-11-23
> **Response to reviewer yUDY (Part 2/2)**
>
> **W2**: Insufficient Emphasis on Contributions.
> > Thank you for your comments. We appreciate the opportunity to clarify the unique contributions of our proposed method.
> >
> > - Firstly, regarding the comparison with Gandelsman et al., as mentioned in Lines 374-375, their work involves decomposing output features and removing spurious features for model editing. However, their approach necessitates prior knowledge about which features are spurious and which heads to remove. This reliance on domain-specific knowledge is a significant limitation. Our contribution addresses this by introducing a data-driven approach to identify and remove spurious features without requiring such prior knowledge. This advancement is detailed in Section 3.2 of our paper.
> > - Secondly, while our method and Santurkar et al.'s method both involve model update for editing, the similarities end there. The objective functions and update processes are fundamentally different. Santurkar et al.'s method requires spatial location information in the representation space for a single image, which is often unavailable and labor-intensive to obtain. In contrast, our method only requires a small set of data with ground-truth labels and predicted error labels, making it more practical and accessible. This distinction highlights our method's contribution to error correction without needing specific domain knowledge.
> > - Lastly, regarding Tip-Adapter, it is indeed a effective method for enhancing CLIP performance in few-shot learning scenarios. However, it is not specifically designed for model editing. While Tip-Adapter can be adapted for CLIP enhancement, it does not provide a solution to address unseen classes during the trarining, which means it can not be served as a model editing method. Therefore, it is not entirely fair to undermine the novelty of our approach simply because both methods are data-driven. Our method's design and application are tailored specifically for effective and localized model editing.
> >
> > In summary, our method offers significant advancements in model editing for CLIP-ViT by eliminating the need for prior knowledge, providing a practical and accessible approach, and ensuring high edit locality. We believe these contributions are novel and valuable to the field.
>
> **W3**: Inconsistent Experimental Design.
>
> > Our method involves a two-phase model editing process. We verify the effectiveness of these two phases using distinct metrics and utilize specific datasets to evalaute distinct metrics:
> > - Phase One: Training-Free Editing
> > This phase focuses on evaluating edit success. We use:
> >
> >    - Waterbirds: This synthetic dataset contains known spurious correlations, making it ideal for comparing our method against Tip-Adapter (data-driven) and TextSpan (leveraging known spurious cues). (Refer to Page 9, Table 1)
> >    - CelebA: This real-world dataset has unknown and latent spurious correlations. Here, we compare our method only to Tip-Adapter, as both are data-driven. (Refer to Page 10, Table 2)
> >
> > - Phase Two: Fine-Grained Editing with Fine-Tuning
> > This phase aims to balance edit success and locality through training. We use:
> >
> >    - Waterbirds + ImageNet-R: We fine-tune the model using 20 Waterbirds samples and test locality on the unseen ImageNet-R dataset. ImageNet-R is a real-world dataset where CLIP performs well. This setup helps us verify if correcting errors in the weakly performing Waterbirds classes affects performance on unrelated, well-performing classes in ImageNet-R.
> >    -  ImageNet-A: As detailed on Page 10, Lines 498–504, we select 10 classes with the most samples, improve performance in the 2 worst-performing classes (edit success), and check if performance remains high on the other 8 unseen classes (locality).
> >
> > - In summary, we use Waterbirds and CelebA in Phase 1 to focus on edit success, and Waterbirds + ImageNet-R and ImageNet-A in Phase Two to balance edit success and locality. CelebA is excluded in Phase Two because its binary classification task (Young vs. Old) lacks the additional classes needed to effectively evaluate locality.
> >
> > We hope this explanation clarifies our experimental design and the rationale behind our dataset choices.

---

> ### Author Response · Authors · 2024-11-23
> **Summary of rebuttal revisions**
>
> We sincerely thank the reviewers for their thoughtful and constructive feedback. Based on the comments, we have made several revisions to improve the clarity, robustness, and comprehensiveness of the manuscript. Below is a summary of the revisions made in response to the reviewers’ suggestions.
>
> **Additional experiments and anlysis**
> >1. Comparison Between Tip-Adapter (Training Version) and Our Training-Free Stage 1 on the Waterbirds Dataset. (Page 9 Table 1)
> >2. More experiments for Our Stage 2 (Page 10 Fig 2)
> >3. Comparison with CLIP-Based Prompt learning approaches (CoOp, CLAP, PLOT) in few-shot scenarios. (Appendix: Page 15 & Table 6)
> >4. Ablation study of $L_{success}(\theta)$ and $L_{locality}(\theta)$ (Appendix: Page 16-17 & Fig 5,Fig 6)
> >5. Sensitivity study of $L_{success}(\theta)$ and $L_{locality}(\theta)$ (Appendix: Page 18-19 & Fig 7,Fig 8)
> >6. Sensitivity analysis for $T$ (Appendix:Page 21 Fig 9)
> >7. Analysis of different update strategy. (Appendix:Page 21 Fig 10, Fig 11, Fig 12)
>
> **Additional visualizations**
> >1. An overview of the proposed method in Page 2. Fig.1.
> >2. More heatmap visualization using Grad-CAM (Appendix: Page 16 Fig 3)
>
> **Other revisions**
> >1. Add callback sentences at the end of Section 3.2 to ensure logical coherence and smooth transition to the subsequent sections.
> >2. More details about how to derive the identification results from the four scores. (Appendix: Page 20 & Algorithm 1)
> >3. An overview of the proposed method  (Appendix: Page 20  Algorithm 2)
> >4. Limitations and directions for future research (Appendix: Page 23)

---

> ### Author Response · Authors · 2024-11-26
> **Gentle Reminder: Follow-up on Rebuttal Response**
>
> Dear reviewer yUDY, we sincerely appreciate your valuable time and effort in reviewing our work. It has been three days since we submitted our response to your concerns, and we would like to kindly confirm whether our replies have effectively addressed your queries. If there are any additional concerns or if further clarifications are needed, please feel free to let us know. We would be more than happy to engage further and address any remaining issues.

---

> > ### Comment · Reviewer_yUDY · 2024-11-26
> >
> > Thank you for your detailed explanations, which have largely addressed my previous concerns. However, I still feel that the current revision of the paper lacks sufficient integration of additional results and reasoning, which affects its overall coherence. As such, I am inclined to maintain my original rating.

---

> ### Author Response · Authors · 2024-11-28
>
> Dear Reviewer yUDY, thank you for your valuable feedback. We appreciate your time and the opportunity to address your concerns. Below, we provide a detailed summary of the experiments and analyses conducted in our paper to highlight the thoroughness of our reasoning and the integration of results:
>
> ### **1. Comprehensive Evaluation Across Datasets**
> We evaluated each stage of our method using both real-world and synthetic datasets to ensure robustness and generalizability:
> - **Stage One:** Evaluated on Waterbirds (synthetic) and CelebA (real-world).  (Section 5.2)
> - **Stage Two:** Evaluated on "Waterbirds + ImageNet-R" (synthetic) and "ImageNet-A" (real-world). (Section 5.3)
>
> We also compared our method with strong baselines, including TextSpan, Tip-Adapter, and Standard Fine-tuning.
>
> ### **2. Reasoning Through Ablation and Sensitivity Analyses**
> To validate the reasoning behind each component of our method, we conducted the following analyses:
> - **Key component Analysis:** Ablation studies and sensitivity analyses were performed on the two critical loss functions $L_{success}(\theta)$ and $L_{locality}(\theta)$ used in Stage Two (see Appendix, pp. 16–19, Figs. 5–8), which
> - **Hyperparameter Analysis:** We analyzed the impact of the hyperparameter $T$, which controls the size of the candidate pool for selecting the final list of attention heads for ablation. (Appendix Page 21, Fig.9)
> - **Updating Strategies:** We thoroughly compared different updating strategies in Stage Two, including full matrix, diagonal matrix, and LoRA.(Appendix:Page 21 Figs 10-12)
> - **Visualization** We provide Grad-CAM heatmap visualizations to illustrate the focus of the ablated attention heads and compare these heatmaps with those generated by TextSpan.(Appendix: Page 16 Fig 3)
> ### **3. Benchmarking Against Other State-of-the-Art Methods**
> - To provide additional insights into our method’s ability to correct errors, we benchmarked it against recent prompt learning methods, such as CoOp, PLOT, and CLAP, in the few-shot CLIP setting. (Appendix: Page 15 & Table 6)
>
> We believe these extensive experiments and analyses showcase the strong integration of results and provide clear reasoning for the effectiveness of our method. We kindly invite you to revisit our findings and reasoning, which are elaborated in both the main text and the appendix.
>
> Thank you once again for your time and thoughtful consideration.

---

### Official Review · Reviewer_AZn1 · 2024-11-04

**Soundness:** 2
**Presentation:** 2
**Contribution:** 3
**Rating:** 6
**Confidence:** 4

**Summary:**

This paper addresses the challenge of correcting CLIP's prediction errors that arise from spurious correlations, particularly in scenarios where only limited data is available and the underlying biases are unknown. The authors propose RefineCLIP, a two-phase model editing framework:

1. In the first phase, the method identifies problematic attention heads through a data-driven approach using four proposed metrics ($\text{DE}_A$, $\text{DE}_B$, $\text{DE}_C$, $\text{DE}_D$) that analyze the direct contributions of attention heads to prediction errors. This is achieved by comparing feature representations between correctly and incorrectly classified samples.
2. In the second phase, the framework introduces a learnable diagonal projection matrix to adapt CLIP's representations. The training objective combines three components:
    - Success loss: aligns model outputs with those of the spurious-feature-ablated model for misclassified samples
    - Locality loss: preserves original predictions for unrelated samples
    - Cross-entropy loss: utilizes available ground truth labels

The authors evaluate their method on both synthetic (Binary Waterbirds) and real-world datasets (CelebA, ImageNet-R, ImageNet-A), demonstrating that RefineCLIP can effectively identify spurious correlations and correct predictions while maintaining locality.

**Strengths:**

- The paper addresses an important and practical scenario where model errors need to be corrected with limited data and without prior knowledge of biases.
- The proposed method for identifying problematic attention heads through data-driven metrics is innovative and well-grounded in the understanding of transformer architectures.
- The experimental design using the Waterbirds dataset with known spurious correlations provides a good validation of the method's ability to identify problematic features.

**Weaknesses:**

## Major weaknesses
1. Limited theoretical foundation
    - The paper heavily relies on Proposition 1 (in appendix) for the equivalence of metrics, but this crucial theoretical foundation is not properly discussed in the main text.
    - The choice of diagonal projection matrix over alternatives (full matrix, LoRA) lacks theoretical justification.
2. Insufficient Comparison
    - The comparison with existing methods is limited mainly to Tip-adapter, ignoring numerous recent CLIP few-shot adaptation methods. For instance, prompt learning approaches like CoOp[^1] have shown strong performance in few-shot scenarios, while more recent methods like PLOT[^2] and CLAP[^3] have further advanced the state-of-the-art in robust few-shot adaptation of vision-language models.
    - The relationship with prompt learning approaches is not discussed, despite potential similarities in goals and methods.

## Minor weaknesses
1. Presentation Issues
    - Complex methodology lacks clear visualizations (e.g., for W, C, A sets and DE metrics calculations)
    - Redundant explanations of symmetric metrics make the paper unnecessarily verbose
2. Limited Analysis
    - The tradeoff between edit success and locality (shown in Fig. 2) deserves more detailed analysis
    - The conclusion lacks discussion of limitations and future directions
    - Hyperparameter sensitivity analysis is inadequate

[^1]: Zhou, Kaiyang, et al. "Learning to prompt for vision-language models." International Journal of Computer Vision 130.9 (2022): 2337-2348.
[^2]: Chen, Guangyi, et al. "Plot: Prompt learning with optimal transport for vision-language models." arXiv preprint arXiv:2210.01253 (2022).
[^3]: Cai, Yichao, et al. "CLAP: Contrastive Learning with Augmented Prompts for Robustness on Pretrained Vision-Language Models." arXiv preprint arXiv:2311.16445 (2023).

**Questions:**

1. Could you clarify why a diagonal projection matrix was chosen over alternatives like full matrix or LoRA? What are the theoretical or practical advantages?
2. How does your method compare with recent prompt learning approaches for few-shot CLIP adaptation? Particularly, methods like CoOp[^1], PLOT[^2], and CLAP[^3] have shown strong performance in similar few-shot scenarios. Could you elaborate on the advantages and disadvantages of your approach compared to these methods?
3. The edit success vs. locality tradeoff seems crucial. Could you provide more insights into how this tradeoff is affected by different factors (e.g., number of samples, choice of target classes)?
4. How sensitive is the method to the choice of hyperparameters α and β in the combined loss function?

[^1]: Zhou, Kaiyang, et al. "Learning to prompt for vision-language models." International Journal of Computer Vision 130.9 (2022): 2337-2348.
[^2]: Chen, Guangyi, et al. "Plot: Prompt learning with optimal transport for vision-language models." arXiv preprint arXiv:2210.01253 (2022).
[^3]: Cai, Yichao, et al. "CLAP: Contrastive Learning with Augmented Prompts for Robustness on Pretrained Vision-Language Models." arXiv preprint arXiv:2311.16445 (2023).

---

> ### Author Response · Authors · 2024-11-23
> **Response to reviewer AZn1 (Part 1/2)**
>
> We appreciate very much your constructive comments on our paper. All revisions made to the paper are highlighted in blue for your ease of reference. We hope that our response satisfactorily addresses the issues you raised.
>
> **Q1 & Major W1.2**: The choice of diagonal projection matrix over alternatives (full matrix, LoRA) lacks theoretical justification.
>
> > We chose a diagonal projection matrix because our method is tailored for scenarios with very limited sample sizes. For example, in the experiments on ImageNet-A, we used just 4 samples per class. Introducing a full matrix would significantly increase the number of trainable parameters, which may cause overfitting given the small sample size.
> >
> > Although LoRA is a low-rank alternative that reduces the number of parameters compared to a full matrix, it is primarily designed for adjusting attention layers within transformers. Our approach, however, keeps all pre-trained model weights fixed and only trains the projection layer, making a diagonal projection matrix a more suitable choice.
> >
> > To futher address the concern, we conducted additional experiments comparing diagonal projections, full matrix projections, and LoRA-style projections. The detailed results and analysis on shown in Appendix: Lines 1123-1170 and Fig.10, Fig.11, Fig.12.
>
> > To summarize, RefineCLIP supports various update strategies, including full matrix, diagonal matrix, and LoRA. The scale of trainable parameters represents a trade-off between fine-tuning capacity and the risk of over-fitting, which can ultimately compromise edit locality. We adopt the diagonal matrix as the standard and representative update strategy in Section 3.3 because it demonstrates stability even in tasks with extremely scarce data while largely preserving edit locality. In contrast, the full matrix  strategy failed completely on ImageNet-A where only 4 samples are available per class.(Fig.11)
>
>
> **Q2 & Major W2**: Comparison with recent prompt learning methods like CoOp, PLOT and CLAP? and relationship with prompt learning approaches**
>
> > Thank you for your suggestion. We have followed your advice to conduct addition experiments to compare our method with three recent prompt learning approaches: CoOp, CLAP, and PLOT. The detailed information is provided in Appendix:Lines 791–800 and Page 17 Table 6. As summarized below, our method performs generally better than all of the other three basedlines on Waterbirds.
>
> |              | Avg. Acc | Wst. Acc |
> |--------------|----------|----------|
> | original CLIP| 80.4%   | 72.3%   |
> | CoOp| **85.7%**   | $\underline{77.7}$%   |
> | Plot| 81.4%   | 71.6%   |
> | CLAP| 81.9%   | 72.3%   |
> | **Ours**| $\underline{85.2}$ %  | **81.0%**  |
>
>
> **Q3. & Minor W2.1** Tradeoff Edit success vs. locality tradeoff regarding different factors (e.g., number of samples, choice of target classes)?
>
> > The trade-off between editing success and locality is primarily influenced by the hyperparameters $\alpha$ and $\beta$. We have conducted additional experiments to evaluate their sensitivity. Please refer to the answer to **Q4 & Minor W2.3** for details.
> >
> > In terms of sample size, we carried out a stability analysis, as shown in Table 3. The results indicate that our method maintains stable editing performance across different sample sizes and sample selections.
> >
> > The choice of target classes for editing is guided by a clear rationale: we select target classes based on the need to correct observed prediction errors within those classes. Our method has been tested on various datasets, encompassing a range of target classes. The consistently strong performance across these datasets underscores the effectiveness of our approach.

---

> ### Author Response · Authors · 2024-11-23
> **Response to reviewer AZn1 (Part 2/2)**
>
> **Q4 & Minor W2.3**: Sensitive analysis for $\alpha$ and $\beta$.
> > Thank you for your advice. We have follow your suggestion to conducted additional experiments to evaluate their sensitivity. The details of these comparisons are presented in Appendix: Lines 856–1018 and Fig.5, Fig.6, Fig.7, Fig.8. In summary, the ablation study demonstrates that $L_\text{locality}(\theta)$ primarily emphasizes locality, while $L_\text{success}(\theta)$ focuses on achieving edit success. Additionally, the sensitivity analysis reveals that when the weights fall between $10^3$ and $10^4$, our method achieves optimal performance in terms of both edit success and locality across the 'Waterbirds+ImageNet-R' and 'ImageNet-A' datasets.
> >
> **Major W1.1**： Proposition 1 is importand but is not well discussed in the main text.
> > We appreciate the reviewer's feedback regarding Proposition 1. In the main text, specifically on Lines 289-291, and further elaborated in Appendix Page 16, we discuss how Proposition 1 demonstrates that $DE_A^{l,h} \approx -DE_B^{l,h}$ and $DE_C^{l,h} \approx -DE_D^{l,h}$. This finding supports the idea that the four scores—$DE_A^{l,h}$, $DE_B^{l,h}$, $DE_C^{l,h}$, and $DE_D^{l,h}$—can be logically grouped into two pairs, reflecting the attention heads they are designed to select. Our experimental results, presented on Lines 410-413 and in Appendix Page 14 Table 4, further substantiate this grouping.
> >
> > While this insight offers a potential reduction in computational complexity, it is not the primary focus of our method. We will enhance its discussion in the main text to clarify its role and implications.
>
> **Minor W1.1**: Visualization for method.
>
> > Thank you for your valuable suggestion. We have followed your advice to include a schematic diagram of the proposed method in Page 2 Fig.1 to enhance readability. Additionally, we have provided a detailed algorithm in Algorithms 1 and 2 to further illustrate the proposed methods. (Appendix: Page 20).
>
>
> **Minor W1.2**: Redundant explanations of symmetric metrics.
>
> > The four metrics discussed on Page 4, Section 3.2 can indeed be grouped into two pairs based on Proposition 1 (Appendix Page 16): $DE_A^{l,h}$ and $DE_B^{l,h}$; $DE_C^{l,h}$ and $DE_D^{l,h}$. This symmetry is supported by Proposition 1 and the experimental results on Lines 410-413. To avoid redundancy, we will streamline the explanation of these metrics in the main text, ensuring clarity while maintaining the necessary context for understanding their symmetry.
>
>
> **Minor W2.2**: Lack limitation and future work.
> >Thank you for your suggestion. We have follow your advice to add a section about the limitation and future work in Appendix (in Page 23).

---

> ### Author Response · Authors · 2024-11-23
> **Summary of rebuttal revisions**
>
> We sincerely thank the reviewers for their thoughtful and constructive feedback. Based on the comments, we have made several revisions to improve the clarity, robustness, and comprehensiveness of the manuscript. Below is a summary of the revisions made in response to the reviewers’ suggestions.
>
> **Additional experiments and anlysis**
> >1. Comparison Between Tip-Adapter (Training Version) and Our Training-Free Stage 1 on the Waterbirds Dataset. (Page 9 Table 1)
> >2. More experiments for Our Stage 2 (Page 10 Fig 2)
> >3. Comparison with CLIP-Based Prompt learning approaches (CoOp, CLAP, PLOT) in few-shot scenarios. (Appendix: Page 15 & Table 6)
> >4. Ablation study of $L_{success}(\theta)$ and $L_{locality}(\theta)$ (Appendix: Page 16-17 & Fig 5,Fig 6)
> >5. Sensitivity study of $L_{success}(\theta)$ and $L_{locality}(\theta)$ (Appendix: Page 18-19 & Fig 7,Fig 8)
> >6. Sensitivity analysis for $T$ (Appendix:Page 21 Fig 9)
> >7. Analysis of different update strategy. (Appendix:Page 21 Fig 10, Fig 11, Fig 12)
>
> **Additional visualizations**
> >1. An overview of the proposed method in Page 2. Fig.1.
> >2. More heatmap visualization using Grad-CAM (Appendix: Page 16 Fig 3)
>
> **Other revisions**
> >1. Add callback sentences at the end of Section 3.2 to ensure logical coherence and smooth transition to the subsequent sections.
> >2. More details about how to derive the identification results from the four scores. (Appendix: Page 20 & Algorithm 1)
> >3. A overview of the proposed method  (Appendix: Page 20  Algorithm 2)
> >4. Limitations and directions for future research (Appendix: Page 23)

---

> > ### Comment · Reviewer_AZn1 · 2024-11-26
> >
> > Thank you for your thorough and detailed response to my review. I appreciate the additional experiments and analyses you have conducted, particularly those comparing your method with recent prompt learning approaches like CoOp, PLOT, and CLAP. The inclusion of visualizations, sensitivity analyses for hyperparameters, and discussions of limitations and future work has enhanced the clarity and robustness of your paper.
> >
> > Based on your revisions, I believe you have adequately addressed the major concerns I raised. Your explanations regarding the choice of a diagonal projection matrix and the trade-off between edit success and locality have provided valuable insights into your methodology. Consequently, I have adjusted my initial rating to reflect these improvements, and I now consider your paper to be above the acceptance threshold.

---

> > > ### Author Response · Authors · 2024-11-26
> > >
> > > Dear Reviewer AZn1, Thank you so much for your thoughtful and encouraging response.
> > >
> > > We are especially grateful for your acknowledgment of our efforts to address your concerns. We deeply appreciate the time and effort you have invested in reviewing our paper and for acknowledging the additional experiments and analyses we conducted.
> > >
> > > Your constructive guidance has significantly contributed to the clarity and robustness of our paper. We are truly honored by your positive evaluation and the adjustment to your rating. Thank you again for your support and thoughtful insights.

---

> ### Author Response · Authors · 2024-11-26
> **Gentle Reminder: Follow-up on Rebuttal Response**
>
> Dear reviewer AZn1,  we sincerely appreciate your valuable time and effort in reviewing our work. It has been three days since we submitted our response to your concerns, and we would like to kindly confirm whether our replies have effectively addressed your queries. If there are any additional concerns or if further clarifications are needed, please feel free to let us know. We would be more than happy to engage further and address any remaining issues.

---

### Official Review · Reviewer_bLQv · 2024-11-10

**Soundness:** 2
**Presentation:** 1
**Contribution:** 3
**Rating:** 5
**Confidence:** 3

**Summary:**

This work focuses on correcting the errors in CLIP-ViT. The authors propose a two-stage model editing framework for this task. In the first stage, they identify which components of the model cause the errors, then nullify these parts to create a spurious-feature-ablated
model that is less influenced by misleading features. In the second stage, the model is edited by learning the error-corrected knowledge for editing and the error-unrelated knowledge from the original model for locality. Experiments are conducted to show the performance of the proposed editing framework.

**Strengths:**

1. The motivation of the work is strong and model editing in CLIP is novel.
2. The proposed causal perspective on error analysis is unique and provides a more targeted way of identifying error sources than conventional feature importance methods.

**Weaknesses:**

1. The title indicates that this work is about CLIP, but this work only focuses on CLIP-ViT. The scopes are inconsistent.
2. The methodology section introduces several techniques, including attention head selection based on causal significance and post-deployment editing, but it is not always clear how these techniques fit together into a cohesive framework. For instance, the logic behind moving from attention head analysis to feature editing may be difficult to follow on a first read.
3. Spurious features can be challenging to separate when entangled with meaningful information.

**Questions:**

1. How does the proposed framework handle cases where spurious correlations are not clear-cut or are entangled with causal features?
2. How is error identification handled differently in synthetic versus real-world datasets, where features are less controlled?
3. Does the method support cases where there are multiple interacting spurious correlations?
4. The two-phase framework includes post-deployment editing, which may increase the overall computational cost. Has the impact on inference speed or computational overhead been measured, especially for large-scale models?
5. How does the model ensure that post-deployment edits improve generalizability rather than just adjusting to recent deployment-specific noise? Are there checks in place to limit overfitting?

---

> ### Author Response · Authors · 2024-11-23
> **Response to reviewer bLQv (Part 1/2)**
>
> We appreciate very much your constructive comments on our paper. All revisions made to the paper are highlighted in blue for your ease of reference. We hope that our response satisfactorily addresses the issues you raised.
>
> **Q1&3**: How to handle spurious correlations that are not clear-cut, or are entangled with causal features, or are multiple interacting.
> > - We understand the reviewer's concern that spurious correlations are implict and complex, which make it challenging to handle. We are conscious of such issues and have stated it in Lines 69-71.
> > - Our approach assumes that model errors are caused by underlying biases, such as spurious correlations. By removing the model's dependence on these biases, we can correct the errors. **Importantly, our method does not require prior knowledge of the underlying biases.** Instead, we identify potential error-causing features using a limited number of erroneously or correctly predicted data points. This allows our method to correct errors regardless of whether the spurious cues are singular, multiple, clear-cut, or intertwined.
> > -  Traditional methods often rely on prior knowledge to identify and handle spurious correlations. However, our proposed method is entirely data-driven and does not require any prior knowledge about these correlations. This is a significant advantage when dealing with hidden or complex spurious correlations.
> > - To further minimize the loss of causal features, we introduce the loss funtion in Eq. (12) that preserves the knowledge from original data in Phase 2 of model training.
> > - Moreover, feature entanglement is often defined at a semantic level. However, our ablation studies operate at the attention head level, which provides a more fine-grained analysis. This means that seemingly entangled features might actually be disentangled at this level, or vice versa. Our Grad-CAM visualizations in Appendix: Page 16, Figure 3, demonstrate that our method can ablate features at a more fine-grained level compared to TextSpan, which relies on prior domain knowledge to remove spurious cues.
>
> **Q2**: Why is error identification handled differently in synthetic versus real-world datasets?
> > We would like to clarify that our approach to error identification is consistent across both synthetic and real-world datasets.
> > - In Phase 1 of our method, we focus on the success of error correction in binary classification tasks. This phase does not require consideration of edit locality, as all data fall within the editing scope. We conduct these experiments on both the Waterbirds dataset (synthetic) and the CelebA dataset (real-world), as detailed in Section 5.2.
> > - We futher evaluate our complete method (with phase 1 and phase 2) in achieving both success editing success and locality using multiple-class classification tasks on the combined 'Waterbirds + ImageNet-R' (a mix of synthetic and real-world data) and ImageNet-A (real-world), as detailed in Section 5.3.
> > - In summary, our method is designed to correct errors effectively across all datasets, regardless of whether they are synthetic or real-world.
>
>
> **Q4**: The computational cost and the inference speed, especially for large-scale models
> > The computational cost of our method is **stable** and **relatively low**.
> >
> > - Our method uses a small number of samples, such as 20 samples for both the Waterbirds and CelebA datasets.
> > - The features for each attention head are not optimized during training and can be precomputed with a single forward pass through the model. This significantly reduces the computational cost. For instance, even with the largest model (ViT-H/14), feature extraction on 20 samples takes less than 20 seconds using an A100 GPU.
> > - With the precomputed feature, the ablated models from Phase 1 can be obtained effectively, taking less than 5 seconds using CPU.
> > - In Phase 2, we only optimize the diagonal matrix, with the number of trainable parameters being equivalent to the feature dimension, typically 768 for CLIP-ViT. This ensures that the computational cost remains low and stable, even for large-scale models.
> >
> > To provide a clearer understanding, we present the training time for **100 epochs** on an A100 GPU for each version of CLIP:
> >
> |      | CLIP-ViT-B/16 | CLIP-ViT-L/14 | CLIP-ViT-H/14 |
> |------|---------------|---------------|---------------|
> | Time | 125 seconds   | 136 seconds   | 207 seconds   |

---

> ### Author Response · Authors · 2024-11-23
> **Response to reviewer bLQv (Part 2/2)**
>
> **Q5**: How does the model ensure that post-deployment edits improve generalizability rather than just adjusting to recent deployment-specific noise? Are there checks in place to limit overfitting?
> > We have implemented a rigorous testing process to test the edit succes and edit locality **using data unseen during training**. The superior empical performance of the proposed method indicates that it does not overfit training data but truly generalizes to unseen data. Concretely,
> > - We have implemented a rigorous testing process:
> >    - Edit Success Testing: We sample data that exhibit the same types of errors as the corrected data, typically from the same classes. This data is **not seen** during model training. By evaluating the model's performance on this unseen data, we can measure how well our edits generalize to new, similar instances.
> >    - Edit Locality Testing: We also sample unrelated data from **different classes** that fall outside the scope of the edits. This data is likewise **unseen** during training. By assessing the model's performance on this unrelated data, we assess whether the edits adversely affect its ability to handle other types of data.
> > - These two metrics—edit success and edit locality—provide a comprehensive evaluation of our model editing method. **The edit success metric demonstrates the generalizability of our corrections, while the edit locality metric ensures that the edits do not cause overfitting or degrade performance on unrelated data**.
>
> **W1**: The title indicates that this work is about CLIP, but this work only focuses on CLIP-ViT.
> > Thank you for your feedback. You are correct that our work specifically focuses on CLIP-ViT. We followed the convention set by previous works, such as Gandelsman et al. (2024), which also used "CLIP" in the title while concentrating on CLIP-ViT. To avoid any confusion, we will revise the title to more accurately reflect the scope of our study.
>
>
> **W2**: It is not clear how these techniques fit together into a cohesive framework. The logic behind moving from attention head analysis to feature editing may be difficult to follow on a first read
>
> > We understand the need for clarity in how our techniques integrate into a cohesive framework. We have summarized our methodology in Lines 98-107 ("Summary of contribution") and Lines 180-185 (beginning of Section 3). We also added a schematic diagram of the proposed method in Page 2 Fig 1, and a detailed algorithm in Algorithms 1 and 2 (Appendix: Page 20) to further illustrate the proposed methods. Here is a detailed breakdown of our framework:
> > - Phase 1: Identifying and Correcting Biases
> >    - Step 1 Identify Biases: We first identify biases in the attention heads that are responsible for errors. This process is detailed in Section 3.2.
> >    - Step 2 Correct Errors: Next, we correct these errors by nullifying the identified biased attention heads using a straightforward zero-ablation strategy. This is explained in Section 3.1.
> > - Phase 2: Achieive edit success and locality
> >    - Step 3 Train Diagonal Projection Matrix: Given that the ablated model from Phase 1 can correct errors but may not maintain edit locality, we train a diagonal projection matrix for the original model to distills error-correcting knowledge from the ablated model for error data while preserving the original model's knowledge for unrelated data. This step is covered in Section 3.3.
> >
> > We first present Step 2 to illustrate how we can correct errors if we identify biased attention heads in our model, and then present Step 1 as an important part of identifying biased attention heads. However, after introducing Step 1 the reader might forget that the role of Step 1 in Step 2.
> >
> > To improve clarity, we will revise the paper to reiterate at the end of Section 3.2 how the head analysis results are used to correct errors. This will help readers better understand the logical flow from attention head analysis to feature editing.
>
>
> **W3**: Spurious features can be challenging to separate when entangled with meaningful information.
>
> > We acknowledge the challenge of separating spurious features from meaningful information. **However**, our approach, as detailed in Q1 & Q3, is entirely data-driven and does not rely on prior domain knowledge. This characteristic allows our method to be applied consistently across various datasets, regardless of the nature of the spurious features—whether they are singular, multiple, clear-cut, or entangled.
> >
> > To validate the effectiveness of our method, we conducted evaluations on real-world datasets such as CelebA and ImageNet-A, where the spurious features are complex and unknown. The results, presented in Table 2 and Figure 2, demonstrate that our approach successfully handles these complexities.

---

> ### Author Response · Authors · 2024-11-23
> **summary of rebuttal revisions**
>
> We sincerely thank the reviewers for their thoughtful and constructive feedback. Based on the comments, we have made several revisions to improve the clarity, robustness, and comprehensiveness of the manuscript. Below is a summary of the revisions made in response to the reviewers’ suggestions.
>
> **Additional experiments and anlysis**
> >1. Comparison Between Tip-Adapter (Training Version) and Our Training-Free Stage 1 on the Waterbirds Dataset. (Page 9 Table 1)
> >2. More experiments for Our Stage 2 (Page 10 Fig 2)
> >3. Comparison with CLIP-Based Prompt learning approaches (CoOp, CLAP, PLOT) in few-shot scenarios. (Appendix: Page 15 & Table 6)
> >4. Ablation study of $L_{success}(\theta)$ and $L_{locality}(\theta)$ (Appendix: Page 16-17 & Fig 5,Fig 6)
> >5. Sensitivity study of $L_{success}(\theta)$ and $L_{locality}(\theta)$ (Appendix: Page 18-19 & Fig 7,Fig 8)
> >6. Sensitivity analysis for $T$ (Appendix:Page 21 Fig 9)
> >7. Analysis of different update strategy. (Appendix:Page 21 Fig 10, Fig 11, Fig 12)
>
> **Additional visualizations**
> >1. An overview of the proposed method in Page 2. Fig.1.
> >2. More heatmap visualization using Grad-CAM (Appendix: Page 16 Fig 3)
>
> **Other revisions**
> >1. Add callback sentences at the end of Section 3.2 to ensure logical coherence and smooth transition to the subsequent sections.
> >2. More details about how to derive the identification results from the four scores. (Appendix: Page 20 & Algorithm 1)
> >3. An overview of the proposed method  (Appendix: Page 20  Algorithm 2)
> >4. Limitations and directions for future research (Appendix: Page 23)

---

> ### Author Response · Authors · 2024-11-26
> **Gentle Reminder: Follow-up on Rebuttal Response**
>
> Dear reviewer bLQv,  we sincerely appreciate your valuable time and effort in reviewing our work. It has been three days since we submitted our response to your concerns, and we would like to kindly confirm whether our replies have effectively addressed your queries. If there are any additional concerns or if further clarifications are needed, please feel free to let us know. We would be more than happy to engage further and address any remaining issues.

---

> ### Comment · Reviewer_bLQv · 2024-12-02
>
> I appreciate the authors' responses to my concerns, most of which have been adequately addressed. However, I still have concerns about the negative impact of editing on the model's performance for unrelated data. Although Figure 2 demonstrates that the proposed method preserves accuracy on other classes better than standard fine-tuning, the accuracy decreases significantly as the number of epochs increases. Furthermore, the baseline standard fine-tuning approach does not include any mechanism to preserve prior knowledge. Therefore, I have decided to maintain my original score.

---

> > ### Author Response · Authors · 2024-12-04
> > **Response to reviewer bLQv**
> >
> > Thank you for your valuable and insightful comments. We understand your concerns regarding the negative impact of editing on the model's performance for unrelated data, which is a well-known issue in model editing. This challenge has been a key motivation for developing our model editing framework.
> >
> > Standard fine-tuning, while effective for achieving edit success, often fails to maintain performance on unrelated data, highlighting the importance of preserving locality. As you noted, there is a trade-off between edit success and edit locality, and our goal is to minimize this trade-off.
> >
> > In Figure 2, our proposed method shows minimal accuracy decrease on the Waterbirds-ImageNet-R combined dataset (second subfigure). However, on the more challenging ImageNet-A dataset (last subfigure), although our method maintains better locality than standard fine-tuning, the accuracy still decreases with more epochs. This issue can be mitigated by implementing early stopping techniques.
> >
> > To the best of our knowledge, before our submission to the conference, there is no feasible model editing method for ViT models. We believe that correcting errors in ViT models is crucial, and our work represents an important initial step in this direction.
> >
> > While our method achieves high edit success with minimal locality loss on the Waterbirds-ImageNet-R combined dataset, it does not perform as well on the ImageNet-A dataset. We are committed to improving our method and believe that further research in the community will lead to better model editing techniques for visual models.

---

### Meta-Review · Area_Chair_XnbR · 2024-12-21

**Metareview:**

This paper proposes to perform modeling editing on a pre-trained CLIP model to remove spurious correlation to improve model performance. The high-level idea is to perform this editing in two stages by first identifying the harmful attention heads and then fine-tuning the model using their proposed loss functions. Experiments demonstrate that their proposed strategy improves model performance and can identify spurious correlations. The paper has several strengths, the reviewers generally have a positive view of the approach and value the importance of the problem. However, the paper's presentation isn't the best. For example, reviewers have concerns over the motivation and the theoretical foundation of the paper and the experiment design. While some of the concerns were addressed during the discussion phase, the reviewers viewed that the paper would need significant changes and the edits did not fully convince them.

**Additional Comments On Reviewer Discussion:**

During the discussion period, the clarifying questions raised by the reviewers were addressed. However, the more significant issues of experiment design and motivation did not convince the reviewers. Furthermore, not all of the responded changes were made to the edited paper. At this point, the paper would need a major revision in writing to incorporate all the materials during the discussion period. The AC encourages the authors to make the changes for a future venue.

---

### Decision · Program_Chairs · 2025-01-22

Reject